# Fish Assemblages as Ecological Indicators in the Büyük Menderes (Great Meander) River, Turkey

**Stamatis Zogaris** [1], **Nicholas Koutsikos** [1,*], **Yorgos Chatzinikolaou** [2], **Saniye Cevher Őzeren** [3], **Kaan Yence** [4], **Vassiliki Vlami** [5], **Pinar Güler Kohlmeier** [6] **and Gürçay Kıvanç Akyildiz** [7]

1  Hellenic Centre for Marine Research, Institute of Marine Biological Resources and Inland Waters, 19013 Anavissos, Greece; zogaris@hcmr.com
2  Independent Researcher, Ioulianou 19, 54351 Thessaloniki, Greece
3  Department of Biology, Faculty of Science, Ankara University, Ankara 06100, Turkey
4  Department of Biology, Faculty of Science, Trakya University, Edirne 22030, Turkey
5  Department of Environmental Engineering, University of Patras, 26504 Rion, Greece
6  Groningen Institute for Evolutionary Life Sciences, University of Groningen, 9712 CP Groningen, The Netherlands
7  Biology Department, Hydrobiology Laboratory, Pamukkale University, Denizli 20160, Turkey
*  Correspondence: nkoutsik@hcmr.gr

**Abstract:** This research describes fish assemblages and associated aquatic ecosystem degradation patterns in the Büyük Menderes River, one of Turkey's most important Anatolian basins. Using standard electrofishing, 44 river sites were sampled throughout the basin accounting for the distribution and abundance of 20 native and seven non-native species, totaling 13,535 fish specimens. At each sampling site, anthropogenic pressures were assessed, and information was gathered to determine the degree of human-induced degradation that may affect fish and their habitats; each site was scored on the basis of a site quality index (SQI). Using the best-available relatively less-degraded river sites, cluster analyses of the samples defined six fish assemblage river types. Further classification of all fish samples utilizing bipartite network analysis resulted in comparable assemblage groupings. The European Fish Index (EFI+) with minor adaptations was applied for assessing river ecological integrity at all sampled sites in order to explore the utility of this widely used index. The EFI+ index results correlated with scores of the SQI but provided a very narrow assessment range, thus failing to accurately and consistently assess the severity of anthropogenic degradation. We recommend a new multimetric index to be developed for the Western Anatolian Ecoregion, of which this basin is a part. The data and insights gained from this exercise may help continue fish-based indicator development for policy-relevant management and conservation in Turkey's rivers.

**Keywords:** bioassessment; river basin; fish; ecological integrity; water framework directive; Turkey





## 1. Introduction

Eastern Mediterranean lotic and associated lentic ecosystems differ from temperate European ecosystems primarily due to biogeographic and climatic conditions, and their biological components show different compositional and structural patterns [1,2]. Balkan and Anatolian river fish assemblages and their species' ecological traits show marked biological differences from Western Mediterranean river basins (e.g., see [3] for species traits; see [4] for endemic assemblages). These differences and distinct assemblages increase the difficulty of transferring and adopting fish-based ecological assessment techniques eastward of Western Europe. Here, we provide research results from a study that explored the distributions and structure of riverine fish communities in a major Anatolian river in order to contribute to policy-relevant river monitoring and conservation applications [5].

Community-level river fish assemblages refer to fish community groupings at particular reaches of water bodies and they reflect the characteristic environmental features of

aquatic ecosystems [6,7]. Fish assemblage types are typically used to describe particular river types documenting and delineating a biotic river typology. These assemblage distribution patterns usually follow a distinct longitudinal gradient reflecting abiotic changes ranging from the upland headwater stream to the lowland delta ecosystems [8,9]. Describing fish assemblages is widely applied in "biotic" river classification, and this is now a classic classification approach (i.e., river type-specific and typology-based) with important applications, particularly where fish are used as indicators of environmental quality for bioassessment and monitoring [7,10]. Community-level bioassessment using fish assemblage data with multimetric indices has expanded since pioneering breakthroughs in the USA, beginning with the publications of James Karr and his colleagues since the early 1980s [11,12]. Karr's index of biotic integrity (IBI), a procedure that numerically depicts associations between human-induced degradation and biological assemblage attributes, has become a standard bioassessment tool widely used in river monitoring and tracking restoration [13].

In most parts of North America, Australia, and Europe, environmental legislation now requires management agencies to adopt or develop biologically-based assessment and monitoring systems for inland waters [8,14]. As fish are large-sized and long-lived consumers, they integrate information on conditions across the food web. Due to their complex ecological requirements, fish in rivers have been proven to be sensitive indicators of ecological integrity; that is, particular aspects of their abundance, species and age-class synthesis, and reproduction reflect the "health" of the river ecosystem [6]. Bioassessment research has shown that fishes are important in indicating many attributes of anthropogenic ecosystem degradation, particularly in rivers and streams [6,15,16]; these include hydromorphological and habitat degradation, water stress and hydrology alteration, longitudinal river continuity fragmentation, severe pollution, and invasive species impacts.

After the year 2000, a revolutionary expansion for fish-based bioassessment took place with the European Union Water Framework Directive (WFD). Fish were promoted as one of four biological quality elements (BQEs) required for routine WFD inland water monitoring in the European Union. The WFD rationale for developing and calibrating fish-based bioassessment in Europe grew rapidly through several major international research projects; an important one was the EU FAME project in the period 2002–2005 [7,10]. However, several countries in Mediterranean Europe initially lacked community-level fish assemblage databases and standardized methods for fish sampling at the time; hence, the development of local indices lagged behind for many years, in some countries [14,17]. In more complex and species-rich river ecosystems, IBI-like statistical indices have developed slowly. Very few have been published in Asia, for example [18]. There are many places which still have not focused on fish-based bioassessment.

In Anatolia, the Asian part of Turkey, work on river fish communities and human-induced pressure–impact analyses has only very recently begun [19–21]. Anatolia is remarkably rich in freshwater biodiversity [22], being a global biodiversity hotspot as shown by the evolutionary patterns of many species of native fishes [23]. Assessing the ecological integrity of rivers and streams is not confined to policy implementation procedures, it should also support biodiversity conservation and ecosystem restoration measures. For these reasons, fish-based bioassessment research should become an important research endeavor in the freshwater ecosystems of Turkey.

In the present study, we surveyed river fish in the Büyük Menderes, a large Turkish river basin. We describe steps taken to explore the patterns and trends in fish distribution and abundance, exploring the potential effects of anthropogenic pressures on sampled fish assemblages. The procedure for fish-based bioassessment follows the methodological framework of the EU WFD; however, there are some premises which complicate the undertaking as there was no fish monitoring background data in this large river basin. The general similarity of the study area's climate type to the neighboring European Union countries in the Mediterranean region provided an opportunity to apply a general model-based bioassessment index, the European Fish Index (EFI+), for the first time in Turkey.

A key aim of our work is to apply the premise of assessing river ecological integrity using the fish assemblages as indicators through on-site sampling and index applications. Ecological integrity is associated with how "pristine" an aquatic ecosystem is relative to the potential or original state of an ecosystem before human-induced degradation. This exploratory application may help guide future developments for much-needed monitoring and conservation management in Anatolia's river basins.

## 2. Materials and Methods

### 2.1. Study Area and Ichthyofaunal Knowledge

In cultural, economic, and ecological terms, the Büyük Menderes is one of the most important river basins in Turkey [24]. The river's tributaries rise in the southwestern Anatolian uplands and flow westward through a wide valley with several steep gorges. The lower river course of the Büyük Menderes expands into a broad, flat-bottomed valley with a typical northeastern Mediterranean landscape near its estuary. Much of the basin, especially in the lowlands, is composed of agricultural land (covering a total of roughly 40% of the basin area) and various semi-natural landforms. The river drains into the Aegean Sea after a course of about 584 km; the total river basin area is approximately 24 873 km$^2$.

The Büyük Menderes is popularly anglicized as the Great Meander [25] and has a long and illustrious cultural history. It has especially been praised since classical Greek times as the Maiandros, a river god. The term "meandering" in geology, literature, art, and architecture originates from this river's ancient name. The river is of outstanding historical significance with prominent historic references to place names in this valley for more than five millennia [26]. More than 30 major ancient cities developed within this basin [27]. The Greek historian Herodotus (484–425 BC) expounded the importance of the river and distinctly mentioned the meandering features of the Nile River by citing the Maiandros as a comparison. The river valley continues to be one of Anatolia's most valuable "breadbaskets", an important agricultural area with substantial industry as well. However, as a result of intensive land-use changes and unregulated human-induced pressures, the riverine ecosystem of the Büyük Menderes basin has been drastically altered, especially during the last half century.

Several recent studies have been conducted to explore the effects of intense pollution and water abstractions on the ecosystems of the Büyük Menderes [24]. Relatively few studies have focused on how the river's degradation effects the biota, such as the fish populations, but there is documentation of histopathological effects on fish from water pollution; heavy metals and other pollutants were much higher than the acceptable limits in many locations [28,29]. With industrial expansion, widespread irrigated agricultural development, and a population of 2.5 million inhabiting the basin, it is widely known that the river basin suffers from a heavy burden of anthropogenic pressures. Furthermore, frequent mention of mass fish kills in the national media has attracted attention to the plight of the river (e.g., internet and news). Despite these pressures, the Büyük Menderes is still an area of outstanding biodiversity values of international renown [25,30].

The Büyük Menderes also plays a role on the international conservation stage with its three internationally important wetlands (namely, Bafa and Işıklı Lakes and the Büyük Menderes Delta) and at least 10 legally protected areas within its basin [24]. The river basin is located within the Western Anatolian Ecoregion [31], a region with exceptionally high fish endemicity, considered a regionally outstanding Freshwater Key Biodiversity Area by the IUCN [22].

Several conservation efforts have begun to target the basin in recent years, yet a satisfactory knowledge of its biodiversity is not complete. During the last 25 years, some of the main tributaries of the Büyük Menderes River's fish fauna have been investigated (e.g., Dipsiz-Çine Creek, İkizdere Creek, and Bafa Lake), but there has not been extensive research on ichthyofaunal assemblages or distributional studies covering the entire river basin [30,32,33]. Güçlü and colleagues [34] published in 2013 a first review of the entire basin's ichthyofuana, representing an excellent baseline survey with 20 sampling stations

including lakes and reservoirs. However, they did not report on historic anthropogenic changes, on the ichthyofaunal community structure, or on some sections of the river, such as the estuarine waters. Güçlü and colleagues [34] proposed that the total species number for the basin amounts to 34 (including six non-native species). Unfortunately, important gaps in taxonomy were observed by these authors, and the taxonomic status of several species was not complete at the time. Some species do not yet have valid names with full consensus by the scientific community (i.e., there are cases where two closely related, wide-ranging species are documented to co-exist in the same basin). Furthermore, the status of some of these species in terms of abundance and frequency of occurrence in the river is still poorly known [34,35]. Many species are scarce, and their longitudinal distributions are uncharted or poorly known. Some fishes such as the sturgeons, *Acipenser* species, are certainly extirpated [36]. There are still some mysteries concerning "missing" fish species, and at least one interviewed local fisher noted that native trout (*Salmo* sp.) may have existed in cold water upland areas; they may now be extirpated from the entire basin. We presume that several undocumented marine fish species enter the lower parts of the river, yet we do not have concrete evidence other than anecdotal information from local fishers during this survey (e.g., this includes unverified statements that *Argyrosomus regius* regularly reproduces in the lower part of the river near its river mouth). A few other species not noted within the river sections were located in adjoining lakes and reservoirs. These presumed deficits in basic natural history inventory are obstacles in monitoring and conservation frameworks [37].

### 2.2. Sampling and Fish Inventory

The sampling was planned within the course of a project aiming to apply the EU WFD monitoring in water bodies in Turkey, funded by an EU project, promoted by the international funding mechanism EuropeAid (see acknowledgements). For the part of the project concerning rivers, sites on several river courses within the Büyük Menderes basin, as well as the main stem of the river, were investigated, primarily focusing on perennial reaches (flowing water all year). Reaches that are currently degraded due to water transfers, engineering projects, and abstractions were also included.

Fish sampling concentrated at the river flow base levels, in the summer–autumn months. Specifically, sampling took place during 31 field days on four separate expeditions: September 2013 and March, April, and June 2014. The March samples were high-flow samples coinciding with heavy rainfall and snowmelt; these were preformed to compare sampling results to base-level flow conditions. The sampling campaigns covered 37 designated river water bodies (WBs), and a total of 44 sites were sampled. Each WB represents a policy-relevant officially delineated river segment. Eleven sites were sampled twice in both high- and low-flow (or base-flow) conditions (Figure 1). All other sites were sampled in low-flow conditions. In total, 55 fish assemblage samples were obtained and entered into a database (MS Excel).

Fish sampling was conducted using a standardized electrofishing technique with the use of a modern backpack electro-fisher, the Smith-Root LR24 980 V. A single anode and the support of three or more persons using dip-nets were employed during sampling. At least four people participated in nearly all samplings. In all samplings, the same team leader (S.Z.) coordinated and participated in all the sampling work, and the same equipment was used in order to maintain consistency across sampling events. The standard group of participants in electrofishing included the coauthors (S.Z., K.Y., V.V., P.G.K., and S.C.O) and two visiting naturalists (see acknowledgements). The work was overseen by Ankara University (S.C.O.). The work team is confident that, in nearly all samplings, the best possible effort was made to keep to a standard of adequately sampling the river reach by including all representative habitats, capturing all fish species present in the river reach, and following the relevant guidelines of the sampling protocol [38].

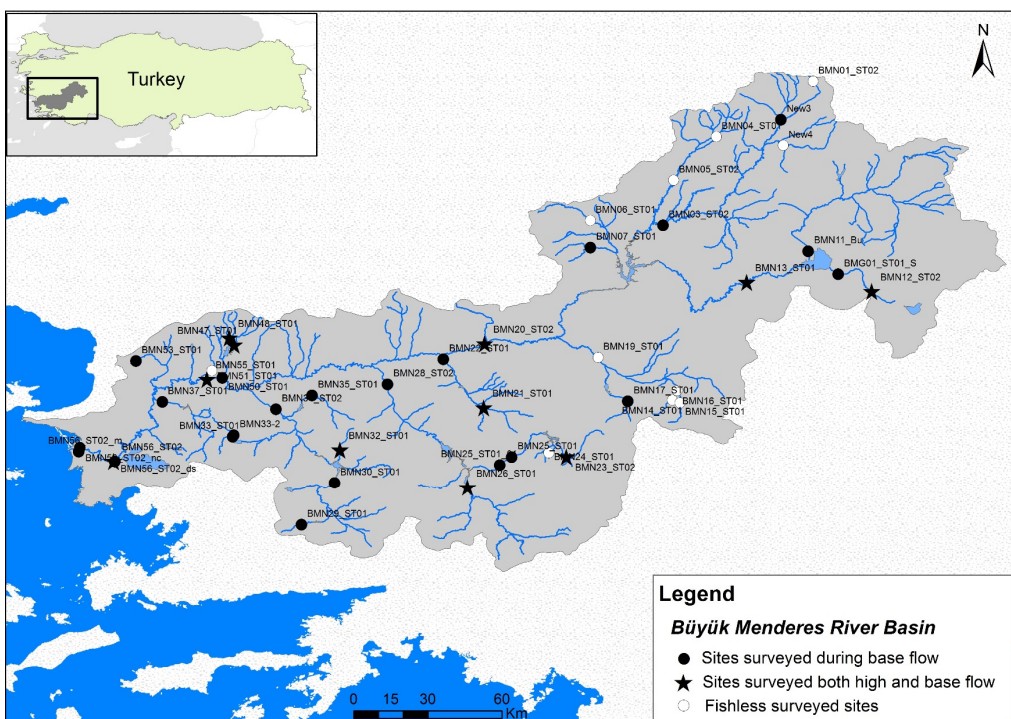

**Figure 1.** Fish sampling sites in the river sections of the Büyük Menderes River basin.

Electrofishing was employed following specifications developed during the EU FAME project where the protocol in this fieldwork was originally developed [17]. This protocol is based on a manual compiled at the Hellenic Center for Marine Research, Institute of Marine Biological Resources and Inland Waters (IMBRIW) [38]. During this EuropeAid project, part of the protocol was also translated into the Turkish language, and a training seminar took place at the Büyük Menderes River. In this project, the recording technique followed a collection and assessment procedure where fish numbers and size-class lengths were documented in a rapid assessment procedure. This approach broadly complies with European CEN standards (see CEN 2003 [39]) in most small wadable rivers and streams investigated. However, it was also necessary to extend sampling in larger river sections with non-wadable areas. In non-wadable deeper river sections extra effort was applied during this project surveys. In such cases, usually more than four people participated using several buckets (collecting fish), and more than three used dip-nets. On three occasions, a boat was used in deeper sections at the lower part of the river. Boats were rented from local fishermen, and the fisher handling the electrofishing generator sat on the bow along with two "netters" on either side. This seemed to be a fairly effective method but could not adequately cover all deeper sections. To complement electrofishing in such challenging reaches, a seine net was also employed in areas where the river was very wide or where there were shallow beach-like habitats. Only one sampling run was conducted at each sampling site, and the longitudinal distance covered was usually 100 m of river stretch (but a minimum of 30 m in small streams, i.e., streams with <5 m wetted channel width). In some sites where fish were not detected during the electrofishing, segment-scale electrofishing continued to search until at least 200 m river length to ensure that the site was fishless (i.e., there was a definite absence of any fish species in the examined river reach). Sites characterized as "fishless" were also presented with a full description of habitat and other parameters in the dataset.

The data collection procedure (fish measurements, etc.) followed the standardized method used for the implementation of the WFD in many Mediterranean countries [38]. In addition, further information about in-stream habitat, anthropogenic pressures, and other attributes of the river site's fish community were collected using the WFD compliant protocol [38]. During most sampling events, fish were not collected for laboratory exami-

nation, a standard regime practice in fish bioassessment monitoring in Europe. Fish were identified on the spot (within the net), and some fish were also photographed in the net and/or using a small portable field aquarium. In some locations, to confirm identifications, fish specimens were collected following the care of experimental animals, consistent with Republic of Turkey animal welfare laws, guidelines, and policies (permission through University of Ankara). Any collected specimens for further lab work were preserved in 5% formaldehyde and later stored in 70% ethanol. Anesthesia using a clove oil solution was performed on fishes in order to photograph fishes in portable aquaria and to euthanize selected specimens for laboratory collection. No biological material was exported out of Turkey following an agreement among researchers at the time. Specimen collection was not the main objective of the sampling; nearly all fishes in the regular sampling events were returned to the river alive.

To complement the site-specific sampling, a species presence review was performed by investigating the available publications and gray literature, and by conducting interviews with knowledgeable researchers (see acknowledgements). We utilized all recent publications to provide an up-to-date taxonomy and used Stout et al. [40] and Vander Laan [41] as references for fish family names. Obviously, gaps in baseline knowledge still exist. Wherever ambiguous or questionable species' distribution data or identification records existed, we clearly stated the suspected record (e.g., we cataloged some unidentified species from our sampling only to genus level).

### 2.3. Statistical Analyses and Fish Assemblage Delineations for River Typology

Simple descriptive statistics and cluster analyses were employed to describe distribution and assemblage patterns and to provide assemblage-based river classification, i.e., a fish-based typology of river water bodies. All fishless sites were excluded from these analyses (10 sites). Unidentified species of marine origin, the genera *Chelon* and *Liza*, were merged together for presentation purposes (see Section 3). Fish species density (ind.·m$^{-2}$) was calculated by using the total individuals of each species divided fished area at each sampling site. Comparing sampling assemblages employed cluster analysis (Euclidean distances and Ward method) on the basis of fish species densities per site and with a fourth root transformation; the cutoff was arbitrarily chosen at a similarity level of 37%.

Species density data were also used in order to apply network analysis. The Gephi Software (v. 0.9.2) was used in order to create and fish species network [42]. Bipartite networks are created between two sets of nodes, and the connections between the nodes are referred to as edges. Connectivity in networks is established entirely through the species they contain [43], and species data, in our case fish densities, are connected through edges with sites, and vice versa [44]. Once the network was created, the "Forced Atlas 2" algorithm was applied for the interpretation of the data [45]. This analysis results in sites that contain common fish species to be closer in the two-dimensional graph. For the discrimination of the different assemblages of the network, a modularity process was used [45], in order to reveal the distinct assemblage type of each fish species modularity. The algorithm was determined with Gephi Software (v. 0.9.2), which also calculates the modularity score that receives values from 0 to 1, where a higher score indicates a more sophisticated internal network structure.

### 2.4. Reference Conditions

Reference conditions in ecological integrity studies refer to the natural or relatively "undisturbed" conditions to be expected at a site or ecosystem type. This is important in order to help construct reference conditions as baselines for the state of optimal quality (i.e., excellent condition) in terms of the structure and function of a biotic community [8,46]. Reference conditions are poorly studied in the Eastern Mediterranean rivers, and few published sources exist for describing biotic reference conditions in river ecosystems in Turkey [19]. Spatially based bioassessment approaches for ecological status may rely on the reference condition approach [47], which involves a comparison of the observed fish

assemblages with a type-specific references (i.e., fish communities in a natural or near-natural state per river type, where available). These conditions are described in relation to geographically delineated river types and water body segments. An EU Twinning project that promoted the initial application of the WFD in the Büyük Menderes basin defined specific river types and water bodies (near-homogeneous river segments) [48]. On the basis of these types and segment delineations, reference conditions should be developed for each water body or modeled for each site.

We chose to explore a reference condition-based biotic typology within this study by identifying assemblage patterns of fishes in certain selected relatively least-disturbed or best available samples. We based our selection of these best-available samples on assessments of the degree of impact of particular anthropogenic pressures per site (see below) as is commonly applied in other fish assessment developments [5,47]. As noted below, some of these best available sites were certainly not near-natural or undisturbed; however, to the best of our knowledge, they provided useful data on community structure in a wide variety of river types.

*2.5. Assessment of Anthropogenic Pressures (On-Site Preclassification)*

The degree of anthropogenic degradation of the sampling sites was assessed according to on-site visual inspection, further data gathered from remote sensing [49], and bibliography relating to each site and/or the broader river section [24]. The methodological format employed here was based on rapid assessment scoring protocols used by trained experts, which have a long history of application in stream and riparian assessments [50,51]. This independent assessment of relative anthropogenic degradation "as it affects fish and habitats" of the sampled sites was accomplished by evaluating and scoring specific anthropogenic pressures that are known to affect fish at each site. This follows premises widely applied in bioassessment development [5,52]. The method applied here developed a simple index, the site quality index (SQI), employing 12 individual pressure elements (anthropogenic degradation criteria) that were scored by the expert assessors at each fish sampling site. Each pressure element (numbered in Table 1) was scored using values of increasing weight: 1 = good condition, 5 = bad condition; thus, when summed, the highest values show the most degraded sites. For example, sites scored as 1, 2, or 3 show no/slight, moderate, or serious alteration, respectively. The score options (Table 1) have different possible score modalities. Some pressure elements were geared to have only three low-score levels (i.e., scores 1, 2, and 3), while proportionally more "weighted" levels were provided where we knew the impact of the specific anthropogenic pressure on fishes/habitats to be especially severe (i.e., the full 1–5 modality provided a more weighted scoring option than 1–3). The sum of the scores of the 12 pressure elements produced the SQI for each site. For more details on such an SQI method within the context of ichthyological index development, see a similar application by Angermeier and Davideanu for Romanian rivers [53].

**Table 1.** The full set of 12 pressure elements, under the seven pressure categories, used for the preliminary fish pressure-based site quality index (SQI). The score modalities and the main assessment method used to evaluate each element are also indicated.

| Anthropogenic Pressure Category | Pressure Element | Score Modalities | Assessment Method |
|---|---|---|---|
| Morphological alteration | 1.Channel alteration | (1, 2, 3, 4, 5) | Visual assessment on-site |
| | 2. Instream/aquatic habitat alteration | (1, 2, 3) | Visual assessment on-site |
| | 3. Embankment restraining riverbed and riparian | (1, 2, 3, 4, 5) | Remote sensing |
| Riparian conditions | 4. Riparian vegetation alteration | (1, 2, 3, 4, 5) | Visual assessment on-site |
| Barriers to fish movement | 5. Barrier upstream—within water body segment | (1, 2, 3) | Remote sensing |
| | 6. Barrier downstream—within water body segment | (1, 2, 5) | Remote sensing |
| | 7. Barrier in the catchment downstream | (1, 3, 5) | Remote sensing |
| Hydrological | 8. Water abstraction affecting site | (1, 3, 5) | Bibliographic references |
| | 9. Hydrological modification of flow regime | (1, 3, 5) | Bibliographic references |
| Hydropeaking | 10. Hydropeaking due to water development, irrigation regulation, and hydroelectric works | (1, 2, 5) | Visual assessment on-site |
| Impounding | 11. Impounding at site and/or segment | (1, 2, 5) | Visual assessment on-site |
| Pollution | 12. Pollution observed during fish and macroinvertebrate sampling or in recent chemical sampling (where available) | (1, 2, 5) | Assessment visually on-site; bibliographic references; physicochemical parameters recorded on-site |

We developed the SQI to summarize information from the on-site anthropogenic pressure scoring. The final SQI analysis resulted in a ranking of all sites along a gradient according to the sum of each site's scores. We chose to categorize the increasing gradient in three classes, i.e., minimal, slight, and severe degradation. Assignment of streams to classes guided by the SQI provides a provisional pressure assessment, a so-called pre-classification, potentially useful for comparative analysis in data-scarce conditions. The use of such "background" pre-classification indices is standard practice in the development of fish-based bioassessment indices, but it has only been recently applied in Eastern Mediterranean rivers [5,54].

*2.6. Fish Index Modification and Application*

Since the sampling work covered a broad range of river degradation conditions, we attempted, in an exploratory manner, to apply the European Fish Index (EFI+) in order to see how this may function in expressing fish-based ecological quality degradation in this river. EFI+ is a model-based index that uses site-specific reference values to calculate reference condition baselines [17]. Site-specific reference values are provided by an undergoing model that predicts the expected reference fish assemblages according to the sampled site's environmental abiotic parameters. The EFI+ model is underpinned by long established stream ecology concepts, within which fish assemblage structure responds to human alterations of aquatic ecosystems in a predictable and quantifiable manner. A central concept for these models is to place each fish in functional trait categories; thus, a limited number of categories (not the species) can be predicted under reference conditions that are known to respond to the different river degradation conditions in a predictable manner. The fish assemblage metrics as provided by the sampled population at each river site should respond to particular environmental variables. Environmental variables required for EFI+ calculation in the Büyük Menderes River were obtained using on-site visual assessment, GIS [55], and remote sensing (see Section 2.5). The EFI+ software requires input data of 12 environmental and sampling parameters: general geological category, river bed sediment size, site altitude, flow regime category, lakes present upstream of site,

upstream drainage area, air temperature in January and July, river slope, site distance from source, river channel wetted width, sampling strategy and method, and fished area. The aforementioned data for each sampled site are provided in Supplementary Table S1.

The European Fish Index (EFI+) was computed after it was adapted for use in this basin where Anatolian endemic species dominate. EFI+ consists of two different fish metrics that vary with general river ecosystem type (salmonid and cyprinid river types). In the Büyük Menderes, applying the EFI+ software is not possible without "fish taxonomic" adaptation due to the many endemic range-restricted species. These Anatolian species are not present in the rivers that were used to develop the index on the European continent. Unfortunately, Anatolian species were not included in the database or software of the EFI+. In our account, a total of 14 out of the 26 species collected were not included in the EFI+ software list [17]. However, since the study area region is within the wider Mediterranean region (with similar climatic, river habitat conditions, and generic fish functional attributes), it was deemed possible to apply the index if we substituted fish names with other ecologically equivalent, related, or functionally similar species present in neighboring European Mediterranean stream systems. Our experience with fish traits and model-based approaches in Greece assisted in this [54]. The basis for adapting the index went by replacing the endemic Anatolian species names with ecologically equivalent species found in Mediterranean Europe. The selection of these "ecological equivalent" fishes to match species traits was based on deductions made from field experience during this sampling survey and expert judgement. Congeneric surrogate species names (i.e., species that are phylogenetically and ecologically similar to the local endemic forms) were used to input data for local and endemic species not considered in the EFI+ software. Supplementary Table S2 shows which species have been replaced with the names adapted. After this adaptation, the EFI+ could run, and all sites were classified according to the fish-based index.

For the EFI+, two subindices are provided; in our application, the Cyprind type subindex was utilized. The Cyprinid type's two metrics are computed as follows:

Ric.RH.Par: richness (number of species in the sample site) of rheophilic species, requiring a rheophilic reproduction habitat, i.e., preference to spawn in running waters.

Ni.LITHO: density (number of individuals per 100 m$^2$ in the sample site) of species requiring lithophilic reproduction habitat, i.e., species which spawn exclusively on gravel, rocks, stones, cobble, or pebbles.

The application is simple:

$$\text{EFI+ Cypr.Fish.Index} = (\text{Ric.RH.Par} + \text{Ni.LITHO})/2.$$

Development of ecological quality ratios (EQRs) requires that each final metric score varies within a finite interval from 0 to 1, and each metric must have the same median value in the absence of any disturbance (i.e., in the undisturbed dataset used to develop EFI+). During the development of EFI+, when only considering undisturbed sites, all four metrics (in both trout and cyprinid ecosystem types) had a median value of 0.80 and very similar values for the 25% quantile (0.69 to 0.73). Table 2 presents the reference baselines and boundaries for standardizing the EQR.

**Table 2.** Summary of the two selected metrics distribution for undisturbed sites for standardizing the EQR.

| Metrics | Min. | 25% Quantile | Median | Mean | 95% Quantile | Max. |
|---|---|---|---|---|---|---|
| Ric.RH.Par | 0.000 | 0.70 | 0.80 | 0.77 | 0.86 | 1.000 |
| Ni.LITHO | 0.000 | 0.71 | 0.80 | 0.73 | 0.83 | 1.000 |

It should be made clear that the underlying premise of using a model-based index such as EFI+ is based on the fact that it may work effectively due to the multifaceted structural and functional effects apparent in ecosystem degradation (and, therefore, trait-based fish assemblage degeneration). As has been said by D.J. Rapport since the 1990s, in terms of the

patterns of anthropogenic degradation, "natural systems, despite their diversity, respond to stress in similar ways". This is a guiding premise in biological monitoring and has also been widely used in other rapid assessment frameworks [56].

## 3. Results

### 3.1. Sampled Sites and Icthyological Results

The investigated stream and river conditions were located at 44 sites (Figure 1, Tables 3 and 4) incorporating 37 water bodies (WBs) (Table 5). These WBs are official management unit river sections as promoted within the EU WFD and were utilized here to show how this work can be of practical management interest. The results show that 10 sites produced fishless samples. Of the fishless sites, two were on intermittent stretches of river that were not flowing during the sampling period.

**Table 3.** Sites sampled in the water bodies (officially delineated river sections) of the Büyük Menderes in this study. The official water body name where the site is located, geographical coordinates, elevation (m a.s.l.), and the number of times the site was sampled are indicated. Where there are two samples, both high- and low-flow conditions (spring-summer) were sampled.

| N | Site Name | Water Body Name | Longitude | Latitude | Elevation | Samples |
|---|---|---|---|---|---|---|
| 1 | BMN01_N3 | Yukari Banaz | 38.747665 | 29.765062 | 916 | 1 |
| 2 | BMN02_N4 | Asagi Banaz1 | 38.654668 | 29.773069 | 945 | 1 |
| 3 | BMN01_ST02 | Yukari Banaz | 38.887430 | 29.882031 | 1264 | 1 |
| 4 | BMN02_ST02 | Asagi Banaz1 | 38.729145 | 29.901113 | 1187 | 1 |
| 5 | BMN03_ST02 | Asagi Banaz2 | 38.363521 | 29.336081 | 537 | 1 |
| 6 | BMN04_ST01 | Dokuzsele1 | 38.686412 | 29.530637 | 899 | 1 |
| 7 | BMN05_ST02 | Dokuzsele 2 | 38.527895 | 29.374914 | 809 | 1 |
| 8 | BMN06_ST01 | Hamam1 | 38.381997 | 29.071666 | 651 | 1 |
| 9 | BMN07_ST01 | Hamam 2 | 38.283191 | 29.072374 | 610 | 1 |
| 10 | BMN11_Bu | Kufi4 | 38.269653 | 29.864130 | 821 | 1 |
| 11 | BMN12_ST02 | Yukari Büyük Menderes 1 | 38.122981 | 30.095312 | 843 | 2 |
| 12 | BMN13_ST01 | Yukari Büyük Menderes 2 | 38.156494 | 29.640245 | 811 | 2 |
| 13 | BMN14_ST01 | Çaykavuştu1 | 37.729445 | 29.370022 | 1019 | 1 |
| 14 | BMN15_ST01 | Çaykavuştu2 | 37.719342 | 29.397376 | 988 | 1 |
| 15 | BMN16_ST01 | Yukarı Çürüksu | 37.763766 | 29.387657 | 860 | 1 |
| 16 | BMN17_ST01 | Gokpinar Deresi | 37.724315 | 29.208474 | 668 | 1 |
| 17 | BMN19_ST01 | Asagi Curuksu2 | 37.884179 | 29.100378 | 179 | 1 |
| 18 | BMN20_ST02 | Orta Büyük Menderes | 37.933321 | 28.687675 | 117 | 2 |
| 19 | BMN21_ST01 | Yukari Dandalaz | 37.700548 | 28.684416 | 451 | 2 |
| 20 | BMN22_ST01 | Asagi Dandalaz | 37.876850 | 28.537744 | 84 | 1 |
| 21 | BMN23_ST02 | Yukari Akcay1 | 37.521739 | 28.985097 | 894 | 2 |
| 22 | BMN24_ST01 | Yukari Akcay2 | 37.537270 | 28.921229 | 894 | 1 |

**Table 3.** *Cont.*

| N | Site Name | Water Body Name | Longitude | Latitude | Elevation | Samples |
|---|-----------|-----------------|-----------|----------|-----------|---------|
| 23 | BMN25_ST01 | Yukari Akcay3 | 37.521221 | 28.785840 | 591 | 1 |
| 24 | BMN26_ST01 | Yukari Akcay4 | 37.408976 | 28.625789 | 319 | 2 |
| 25 | BMN28_ST02 | Asagi Akcay | 37.786307 | 28.334788 | 60 | 1 |
| 26 | BMN29_ST01 | Girme Deresi | 37.273232 | 28.021417 | 499 | 1 |
| 27 | BMN30_ST01 | Yukari Cine1 | 37.426505 | 28.141515 | 266 | 1 |
| 28 | BMN32_ST01 | Yukari Cine3 | 37.547058 | 28.161884 | 556 | 2 |
| 29 | BMN33_ST01 | Asagi Cine1 | 37.595376 | 27.771665 | 296 | 1 |
| 30 | BMN33-2 | Asagi Cine1 | 37.601343 | 27.775746 | 310 | 1 |
| 31 | BMN34_ST02 | Asagi Cine2 | 37.695438 | 27.928781 | 40 | 1 |
| 32 | BMN35_ST01 | Asagi Cine3 | 37.745620 | 28.060388 | 140 | 1 |
| 33 | BMN37_ST01 | Asagi Saricay | 37.722100 | 27.515985 | 11 | 1 |
| 34 | BMN47_ST01 | Yukari Ikizdere1 | 37.928269 | 27.777246 | 195 | 2 |
| 35 | BMN48_ST01 | Yukari Ikizdere2 | 37.954009 | 27.758918 | 216 | 2 |
| 36 | BMN50_ST01 | Asagi Ikizdere2 | 37.809497 | 27.734028 | 23 | 1 |
| 37 | BMN51_ST01 | Yalki | 37.836625 | 27.695994 | 24 | 1 |
| 38 | BMN53_ST01 | Naipli Cayi | 37.870638 | 27.419755 | 164 | 1 |
| 39 | BMN55_ST01 | Asagi Büyük Menderes1 | 37.803416 | 27.677998 | 21 | 2 |
| 40 | BMN56_ST02 | Asagi Büyük Menderes2 | 37.505351 | 27.337874 | 3 | 2 |
| 41 | BMN56_ST02_ds | Asagi Büyük Menderes2 | 37.505606 | 27.342842 | 4 | 1 |
| 42 | BMN25_ST01_01 | Yukari Akcay3 | 37.492186 | 28.742547 | 541 | 1 |
| 43 | BMN56_ST02_m | Asagi Büyük Menderes2 | 37.541314 | 27.211556 | 0 | 1 |
| 44 | BMN56_ST02_nc | Asagi Büyük Menderes2 | 37.555730 | 27.215484 | 1 | 1 |

At total of 20 native and seven non-native (alien) taxa were confirmed within the river sections in our study (Table 4). The nomenclature of our species list was curated with a review of all available literature (i.e., [34,57,58]). Expert review of the list was also provided through the gracious assistance of local experts (see acknowledgements), but there were still challenges in taxonomy and/or sampled specimen identifications. Some specimens were not identified to species level in the field. If there was the likelihood of any conceivable doubt in identification, only the genus name or a provisional nomenclature, a so-called operational taxonomic unit (OTU), is given. In some circumstances, these operational names may include two similar-looking species of the same genus. An example is the very widespread chub, which was kept to genus level ("*Squalius* sp.") during field identification while sampling. Published documentation refers to two valid chub species in the basin [34], *S. fellowesi* and *S. carinus*; however, since these were not consistently distinguishable in the field, we accounted for only one OTU which we referred to as "*Squalius* sp." during sampling and as *Squalius fellowesi/carinus* in our final list. For the same reason the local nase species is presented as *Chondrostoma turnai/meandrense*. These taxonomic presentations were required because different species names and a changing taxonomy has been evolving

in this basin in recent years [34,59]. Since molecular confirmation was not possible during this rapid sampling bioassessment project, extra care was taken to document what was collected. For presentation purposes in this paper, each species/OTU was given a short name code (Table 4) and distributional documentation is provided at the official water body level (Table 5). Every taxon was photographed on a net, in the water, or within small field aquaria to document species finds (e.g., Figure 2).

**Table 4.** Species collection summary. The local taxonomic reference refers to confirmation and the most recent reference to the species presence in the Büyük Menderes Basin. The catch per unit effort (CPUE) is simply related to the detected density based on individuals caught per area sampled.

| Taxon | Species Code | Local Taxonomic Reference | n | F.O. (%) | Mean Abundance (ind.) | CPUE (ind./m$^2$) |
|---|---|---|---|---|---|---|
| *Native species* | | | | | | |
| *Alburnoides smyrnae* (Leuciscidae) | *Alsm* | [60] | 21 | 3.64 | 0.38 | 0.0022 |
| *Alburnus demiri* (Leuciscidae) | *Alde* | [60] | 80 | 18.18 | 1.45 | 0.0035 |
| *Anatolichthys maeandricus* (Aphaniidae) | *Apme* | [61] | 1 | 1.82 | 0.02 | 0.0000 |
| *Barbus xanthos* (Cyprinidae) | *Bape* | [34] | 1193 | 43.64 | 21.69 | 0.0737 |
| *Capoeta aydinensis* (Cyprinidae) | *Cabe* | [62] | 148 | 16.36 | 2.69 | 0.0069 |
| *Chondrostoma turnai/meandrense* (Leuciscidae) | *Chme* | [20] [63] | 1046 | 30.91 | 19.02 | 0.0865 |
| *Cobitis afifeae* (Cobitidae) | *Cofa* | [34] [64] | 191 | 20.00 | 3.47 | 0.0099 |
| Unidentified *Cyprinidae* | *Cyprsp* | | 7 | 1.82 | 0.10 | 0.0003 |
| *Dicentrarchus labrax* (Moronidae) | *Dila* | [34] | 6 | 3,62 | 0.1 | 0.0004 |
| *Garra menderesensis* (Cyprinidae) | *Gaki* | [65] [35] | 1 | 1.00 | 0.12 | 0.0001 |
| *Gobio maeandricus* (Gobionidae) | *Gome* | [34] | 19 | 7.27 | 0.35 | 0.0006 |
| *Knipowitschia caucasica* (Gobiidae) | *Knca* | [34] | 5 | 1.82 | 0.09 | 0.0004 |
| *Chelon labrosus* (Mugilidae) | *Chela* | | 6 | 3.64 | 0.11 | 0.0009 |
| *Liza* spp. (Mugilidae) | *Liza1* | | 88 | 5.45 | 1.60 | 0.0043 |
| | *Liza2* | | 29 | 5.45 | 0.53 | 0.0015 |
| *Mugil cephalus* (Mugilidae) | *Muce* | [34] | 639 | 5.45 | 11.62 | 0.0450 |
| *Luciobarbus kottelati* (Cyprinidae) | *Luko* | [66] | 300 | 23.64 | 5.45 | 0.0122 |
| *Oxynoemacheilus germencicus* (Nemacheilidae) | *Oxynsp* | [34] [35] | 3589 | 50.91 | 65.25 | 0.2747 |
| *Petroleuciscus ninae* (Cyprinidae) | *Pesm* | [66] | 90 | 14.55 | 1.64 | 0.0094 |
| *Squalius fellowesi/carinus* (Leuciscidae) | *Squasp* | [67] | 3608 | 54.55 | 65.60 | 0.2249 |
| *Vimba mirabilis* (Leuciscidae) | *Vimi* | [34] | 353 | 20.00 | 6.42 | 0.0127 |
| *Non-native species* | | | | | | |
| *Carassius gibellio* (Cyprinidae) | *Cagi* | [34] | 779 | 27.27 | 14.16 | 0.0705 |
| *Cyprinus carpio* (Cyprinidae) | *Cyca* | [34] | 2 | 3.64 | 0.04 | 0.0001 |
| *Gambusia holbrooki* (Poeciliidae) | *Gaho* | [34] | 837 | 20.00 | 15.22 | 0.1253 |
| *Lepomis gibbosus* (Centrarchidae) | *Legi* | [34] | 266 | 20.00 | 4.84 | 0.0215 |
| *Pseudorasbora parva* (Gobionidae) | *Pspa* | [34] | 7 | 7.27 | 0.13 | 0.0009 |
| *Rhodeus amarus* (Acheilognathidae) | *Rham* | [34] | 105 | 7.27 | 1.91 | 0.0044 |
| *Tinca tinca* (Tincidae) | *Titi* | [34] | 119 | 5.45 | 2.16 | 0.0037 |

**Table 5.** Species collected per water body at all river sampling sites in this project (species in code form; see Table 4). Official water body names with specific sampling site locations in Table 3.

| N | Water Body Name | No. Sites | No. Samples | No. Species Recorded | Species Recorded |
|---|---|---|---|---|---|
| 1 | Aşağı Sarıçay | 1 | 1 | 4 | *Cagi, Cofa, Gaho, Pesm* |
| 2 | Aşağı Çine1 | 2 | 2 | 2 | *Pesm, Squasp* |
| 3 | Dokuzsele-2 | 1 | 1 | 0 | *FISHLESS* |
| 4 | Hamam2 | 1 | 1 | 2 | *Oxynsp, Squasp* |
| 5 | Aşağı Çürüksu2 | 1 | 1 | 0 | *FISHLESS* |
| 6 | Gökpınar Deresi | 1 | 1 | 1 | *Oxynsp* |
| 7 | Yukarı Akçay1 | 1 | 2 | 6 | *Bape, Cagi, Legi, Oxynsp, Squasp, Alde* |
| 8 | Yukarı Dandalaz | 1 | 2 | 4 | *Bape, Cabe, Oxynsp, Squasp* |
| 9 | Aşağı Akçay | 1 | 1 | 12 | *Alde, Bape, Cabe, Chme, Cofa, Cyprsp, Gaho, Luko, Oxynsp, Pesm, Squasp, Vimi* |
| 10 | Aşağı Çine2 | 1 | 1 | 10 | *Alde, Cagi, Chme, Cofa, Legi, Luko, Oxynsp, Pesm, Squasp, Vimi* |
| 11 | Girme Deresi | 1 | 1 | 2 | *Bape, Squasp* |
| 12 | Yukarı Çine1 | 1 | 1 | 6 | *Alsm, Bape, Cabe, Luko, Oxynsp, Squasp* |
| 13 | Aşağı Dandalaz | 1 | 1 | 3 | *Cofa, Oxynsp, Squasp* |
| 14 | Aşağı Çine3 | 1 | 1 | 3 | *Cabe, Legi, Squasp* |
| 15 | Yukarı Akçay2 | 1 | 1 | 0 | *FISHLESS* |
| 16 | Çaykavuştu2 | 1 | 1 | 0 | *FISHLESS* |
| 17 | Çaykavuştu1 | 1 | 1 | 0 | *FISHLESS* |
| 18 | Yukarı Çürüksu | 1 | 1 | 0 | *FISHLESS* |
| 19 | Kufi4 | 1 | 1 | 9 | *Vimi, Bape, Cagi, Chme, Gome, Squasp, Titi, Gaki, Apme* |
| 20 | Yukari Akcay4 | 2 | 2 | 8 | *Alde, Bape, Cabe, Chme, Luko, Oxynsp, Squasp, Vimi* |
| 21 | Yukari Cine3 | 1 | 2 | 1 | *Bape* |
| 22 | Yukari Ikizdere2 | 1 | 2 | 4 | *Bape, Cagi, Oxynsp, Squasp* |
| 23 | Hamam1 | 1 | 1 | 0 | *FISHLESS* |
| 24 | Asagi Büyük Menderes1 | 1 | 2 | 13 | *Alde, Cagi, Chme, Cofa, Cyca, Gaho, Legi, Luko, Oxynsp, Pesm, Pspa, Rham, Vimi* |
| 25 | Yukarı Banaz | 2 | 2 | 3 | *Bape, Oxynsp, Squasp* |
| 26 | Aşağı Banaz1 | 2 | 2 | 0 | *FISHLESS* |
| 27 | Yukari Büyük Menderes 1 | 1 | 2 | 8 | *Cagi, Chme, Cofa, Gome, Luko, Oxynsp, Squasp, Titi* |
| 28 | Asagi Banaz2 | 1 | 1 | 6 | *Alsm, Bape, Cagi, Oxynsp, Squasp, Cyprsp* |
| 29 | Yukari Akcay3 | 2 | 2 | 5 | *Bape, Cabe, Chme, Oxynsp, Squasp* |
| 30 | Asagi Büyük Menderes2 | 4 | 5 | 15 | *Alde, Cagi, Chme, Gaho, Legi, Chela, Muce, Vimi, Pspa, Rham, Dila, Liza1, Liza2, Cyca, Knca* |
| 31 | Yalkı | 1 | 1 | 0 | *FISHLESS* |
| 32 | Asagi Ikizdere2 | 1 | 1 | 8 | *Alde, Cagi, Chme, Gaho, Luko, Pesm, Rham, Vimi* |

| N | Water Body Name | No. Sites | No. Samples | No. Species Recorded | Species Recorded |
|---|---|---|---|---|---|
| 33 | Naipli Cayi | 1 | 1 | 4 | *Bape, Cabe, Oxynsp, Squasp* |
| 34 | Yukari Ikizdere1 | 1 | 2 | 3 | *Bape, Oxynsp, Squasp* |
| 35 | Yukari Büyük Menderes 2 | 1 | 2 | 9 | *Bape, Cagi, Chme, Cofa, Gaho, Gome, Luko, Oxynsp, Squasp* |
| 36 | Orta Büyük Menderes | 1 | 2 | 8 | *Alde, Bape, Chme, Cofa, Luko, Oxynsp, Squasp, Vimi* |
| 37 | Dokuzsele-1 | 1 | 1 | 0 | *FISHLESS* |

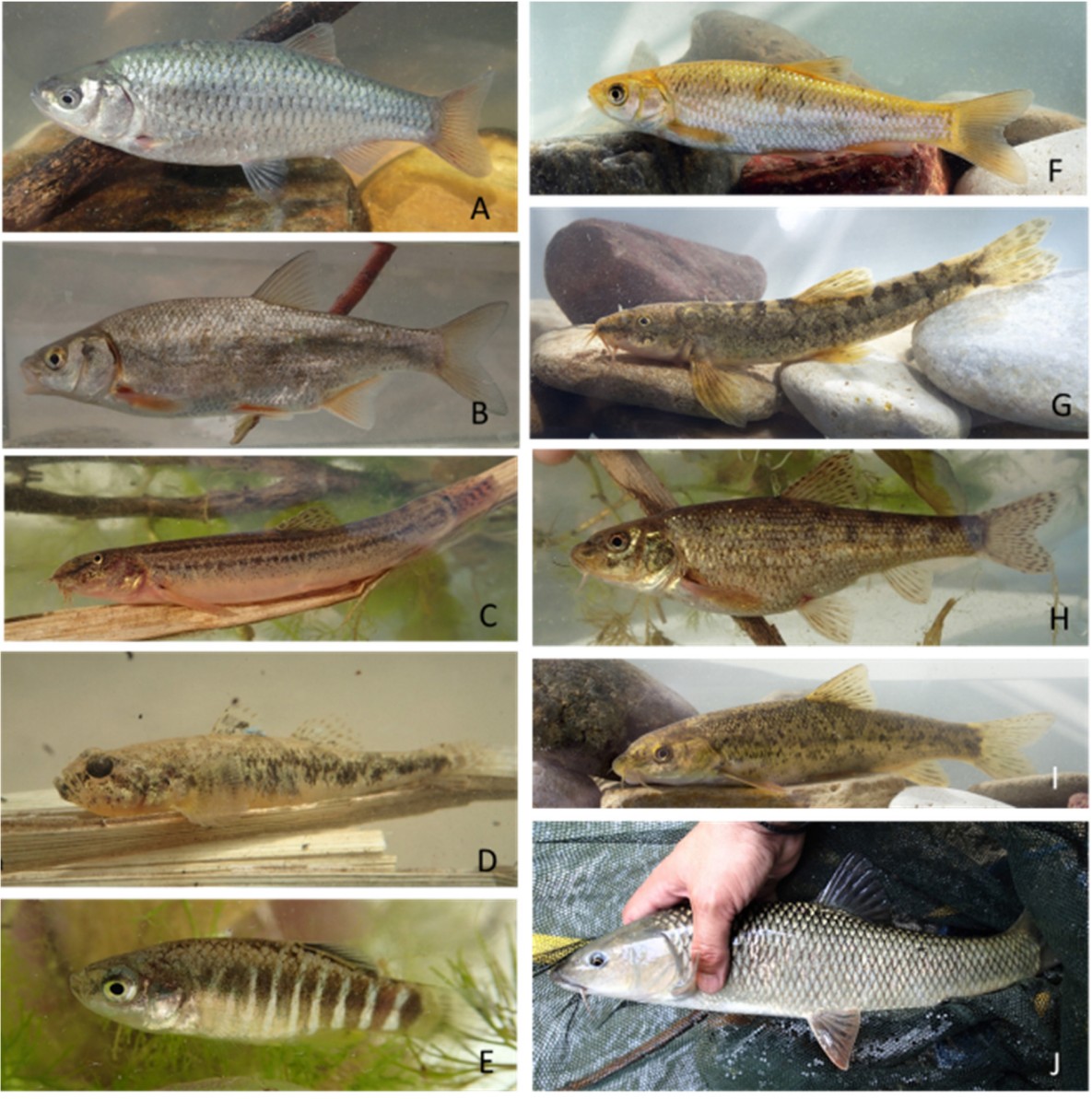

**Figure 2.** Ten characteristic native fish species collected during the 2013–2014 survey: (**A**) *Petroleuciscus ninae*, (**B**) *Vimba mirabilis*, (**C**) *Cobitis afifeae*, (**D**) *Knipowitschia caucasica*, (**E**) *Anatolichthys maeandricus*, (**F**) *Squalius fellowesi/carinus*, (**G**) *Oxynoemacheilus germenicus*, (**H**) *Gobio maeandricus*, (**I**) *Barbus xanthos*, and (**J**) *Luciobarbus kottelati*. Fish sizes are not to scale; photographed on site by S. Zogaris and A. Vidalis.

The most abundant native taxa were *Squalius fellowesi/carinus*, *Oxynoemacheilus germencicus*, *Barbus xanthus*, and *Chondrostoma turnai/meandrense*, all being range-restricted western Anatolian endemics. In terms of the species richness and abundance it is important to note that invasive alien species such as *Carassius gibelio*, *Gambusia holbrooki*, and *Lepomis gibbosus* contributed with a fairly high mean abundance (i.e., 14.1%, 15.2%, and 4.8%, respectively) (Table 4).

### 3.2. Environmental Assessment Results

The SQI represents a provisional assessment of perceivable on-site and river segment pressures that may seriously affect the ichthyofauna and their habitats at the specific sampling sites (Figure 3).

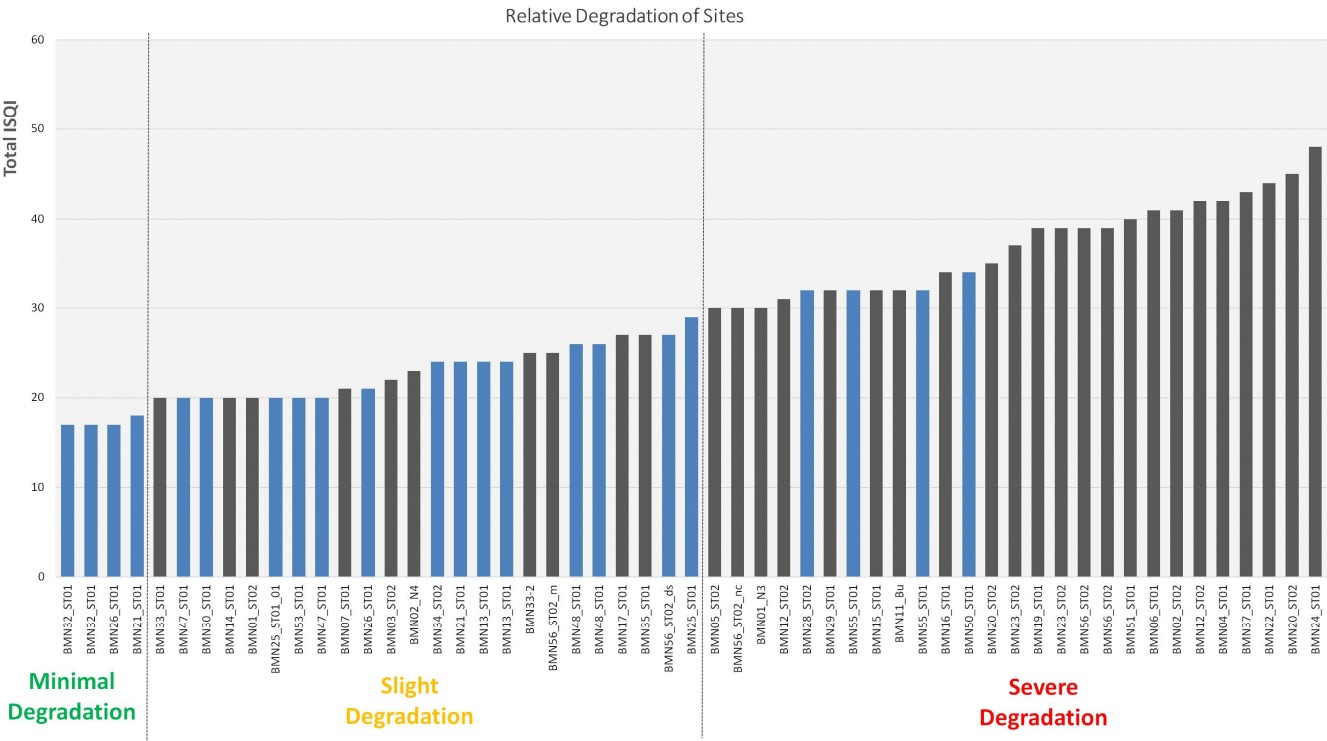

**Figure 3.** The site quality index (SQI) presents a gradient of degradation of all samples in the Büyük Menderes River Basin using the on-site pressure assessment method (pre-classification). The samples include the full breadth of ichthyological river samples in the basin. These are shown as bars with an increasing degree of anthropogenic degradation and categorized in three degradation classes. Boundaries of the degradation classes (minimal, slight, and severe) are designated arbitrarily at selected step-changes along the degradation gradient (i.e., as SQI scores progressively increase along the histogram). The 22 river sites, in blue, refer to those chosen (by expert judgment) to represent "best available references" for constructing a preliminary biotic river typology.

According to the generic three-category degree of degradation (minimal, slight, severe) with the application of the site-assessed SQI, a thematic map allows us to show the gradient of degradation as assessed on-site in a pre-classification of degradation, i.e., before fish assemblages were analyzed (Figure 4). As expected, the summed weighting denoted the pressure parameters in a mechanistic manner, and most sites were characterized as being in poor condition (slight or severe degradation).

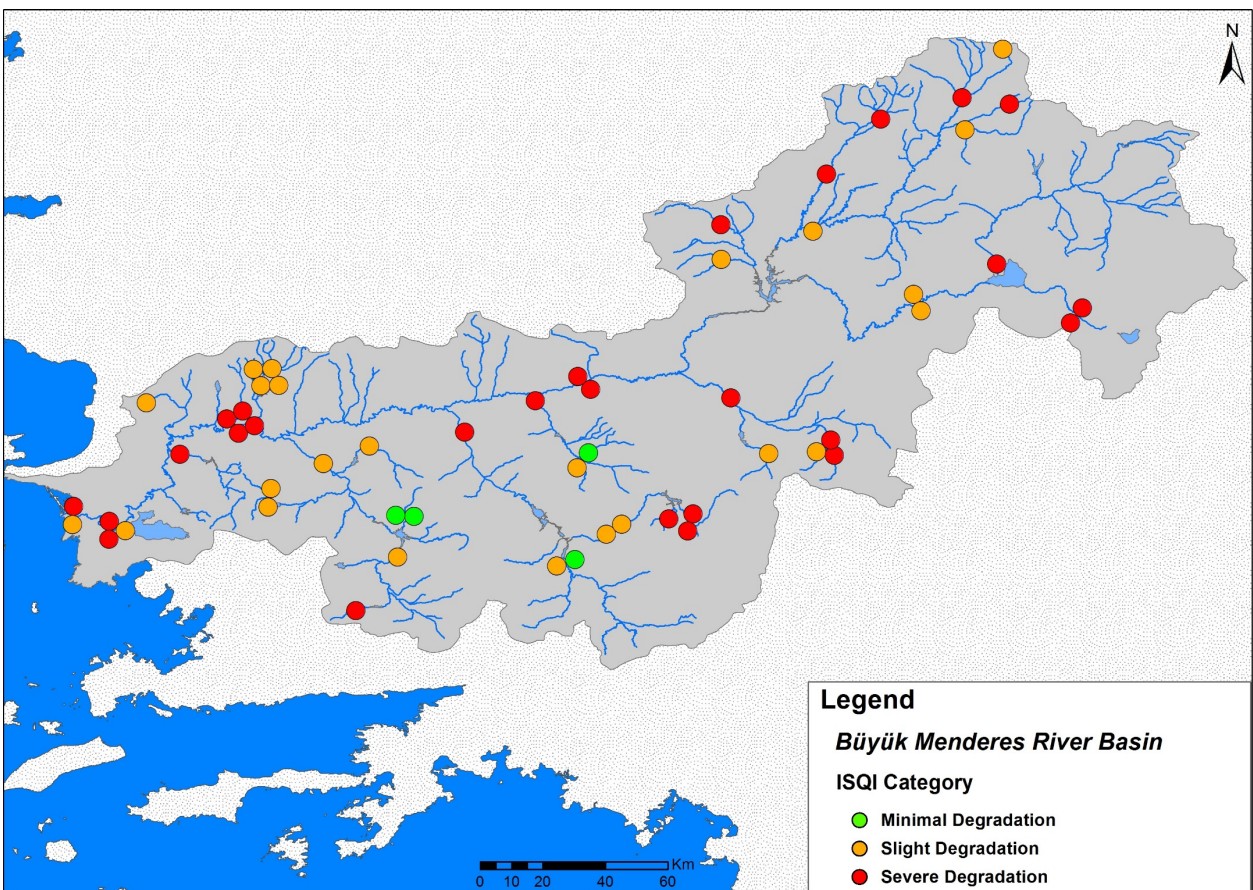

**Figure 4.** The degradation classes (minimal, slight, and severe) of all samples (n = 55) defined with the application of the site quality index (SQI) based on expert-defined step-change cutoffs (see Figure 3).

### 3.3. Biotic Typology Based on Fish Assemblage Types

The SQI application showed that very few sites could be assessed as proper "reference sites" (i.e., near-natural) on the basis of WFD-compliant research procedures. The least impacted sites (referring to the SQI score) and some selected best-available sites were utilized in the cluster analyses of the fish assemblages (Figure 5) to provisionally define fish assemblage river types. These biotic river types were delineated and mapped based on the river length delineations of the national water bodies (Figure 6). Six generic biotic river types were provisionally defined as follows:

o SU = small upland (all small and very small tributaries);
o SL = small lowland (small low-elevation main tributaries);
o LU = large upland (mid-section main stem and two major low-land tributaries);
o LL = large lowland;
o LM = large main stem (the major meandering lower section of the river's main stem);
o LD = large delta (large channel reaches with free communication with the sea and surrounding lagoonal wetlands).

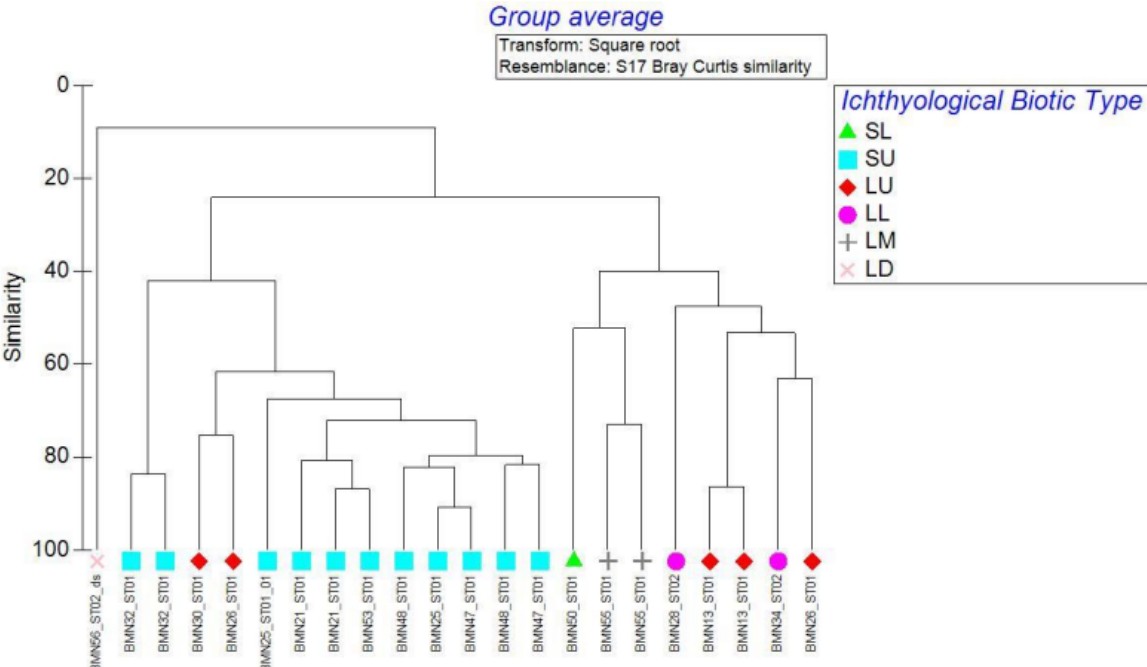

**Figure 5.** Twenty-two selected samples showing "best available" higher-quality conditions (based on SQI assessment; Figure 3) classified using cluster analysis (fish assemblage and species density). This classification assisted in the definitions of six ichthyological biotic types for the surveyed water bodies (see text below).

Bipartite network analysis was used to elucidate assemblage patterns and compare with the provisional classification of biotic types. Gephi Software (v. 0.9.2) generated the matrix between sampling sites and fish species densities. The resulting data had 61 nodes and 184 edges. The Forced Atlas 2 algorithm applied to the bipartite matrix and the graphical representation of the network illustrated one large interconnected structure (Figure 7a). Once the network was created, the implementation of the community detection algorithm (modularity test = 0.555) generated seven modularity classes. Overall, five of the modularity classes enclosed over 90% of the total nodes, while each one of the remaining two classes displayed percentages below 5%. Since all samples were used (not just 22), the seven network groups were comparable to the six main biotic river types (Figure 7b).

The bipartite network analysis and biotic classification developed through the cluster analyses both allude to a gradient of assemblage changes along an upland–lowland axis (evident in Figure 6). Moreover, fish composition and abundance differed markedly on the basis of river size and relation to the main stem. The two largest fish assemblages were modularity class 4 (24.59%) with six sites (large river sites, main stem), and class 3 (21.31%) with eight sites (small river sites, tributaries). Class 4 included nine species with *Chondrostoma*, *Vimba*, and *Luciobarbus* being the most dominant genera, while class 3 contained five species, with *Squalius* being the dominant genus. The other two largest modularity classes were class 5 and class 6 (equally 16.39%) with six sites (distinctly small upland rivers) and three sites (distinctly lowland large deltaic river), respectively. *Barbus* and *Capoeta* were the dominant genera in class 5, while class 6 had the distinctive species of the estuaries and delta lagoons (*Chelon/Liza*, *Mugil*, and *Dicentrarchus* genera). The remaining assemblages displayed were class 1 (11.48%; six sites in small rivers both upland and lowland), class 2 (4.92%; one lowland wetland site), and class 0 (4.92%; two small river sites), with *Oxynoemacheilus*, *Cobitis*, and *Petroleuciscus* being the dominant genera, respectively (Figure 7). Of course, due to local habitat conditions, it is normal to see a divergence into various localized fish assemblages, as recorded for the last three modules.

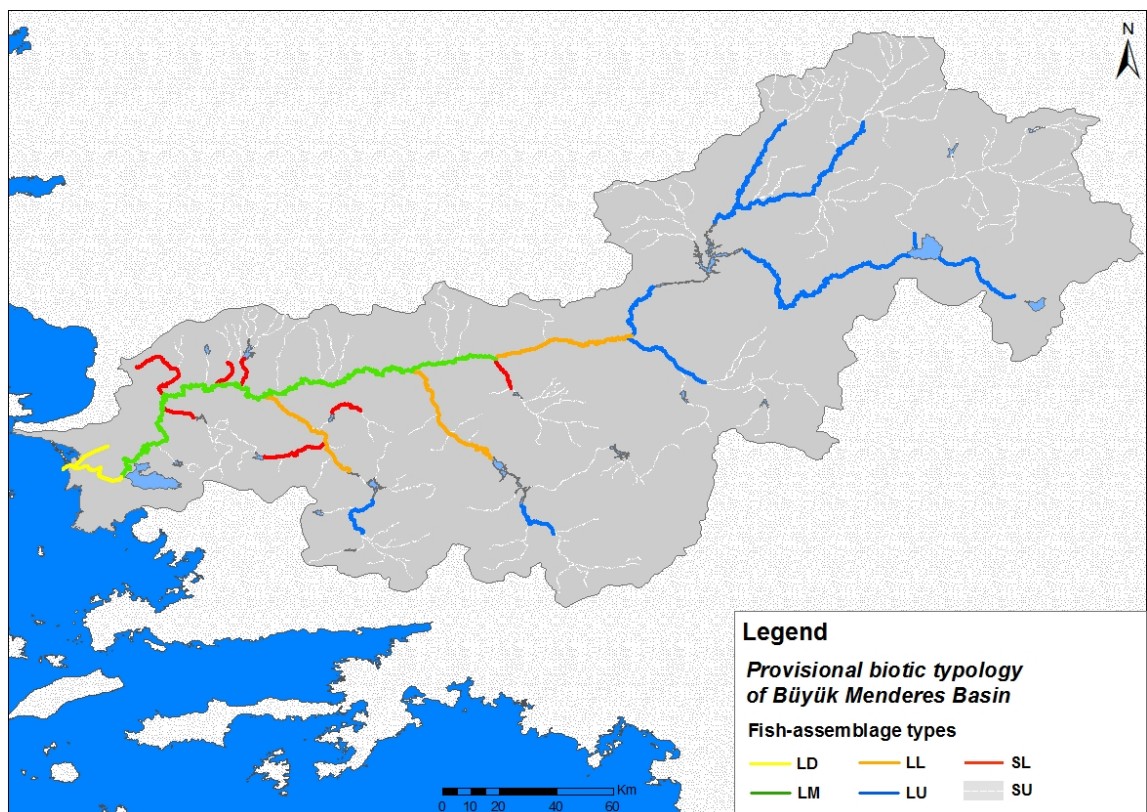

**Figure 6.** Provisional biotic typology based on fish assemblage types where sampling was deemed sufficient to classify the river's official water bodies (WBs) in color. River types codes are: SU = small upland and all very small tributaries; SL = small lowland; LU = large upland; LL = large lowland; LM = large main stem; LD = large delta. This pattern of community assemblages was also complemented by bipartite network analysis (see Figure 7).

In the bipartite network analysis, all recorded samples (from all sampled sites) were used, while river types were developed on the basis of only 22 selected "reference" samples including some hand-picked as "best available" but still degraded. The larger sample set (55 samples) obviously had the potential to produce idiosyncrasies due to widespread degradation and highly disturbed/depauperate assemblage samples. Furthermore, some scarce species produce idiosyncratic groupings. In Figure 7, this was the case with the presence of *Anatolichthys*, which was only found in a slow-flowing upland area near Işıklı lake, and *Cyprinus*, which was only captured in the lowest reaches of the delta.

### 3.4. Application of the European Fish Index (EFI+)

The bioassessment of ecological integrity based on the fish samples using the EFI+ provides a generic model-based approach to building site-specific references. The results of the EFI+ index application, providing an EU WFD standard five-class quality status classification, are mapped in Figure 8.

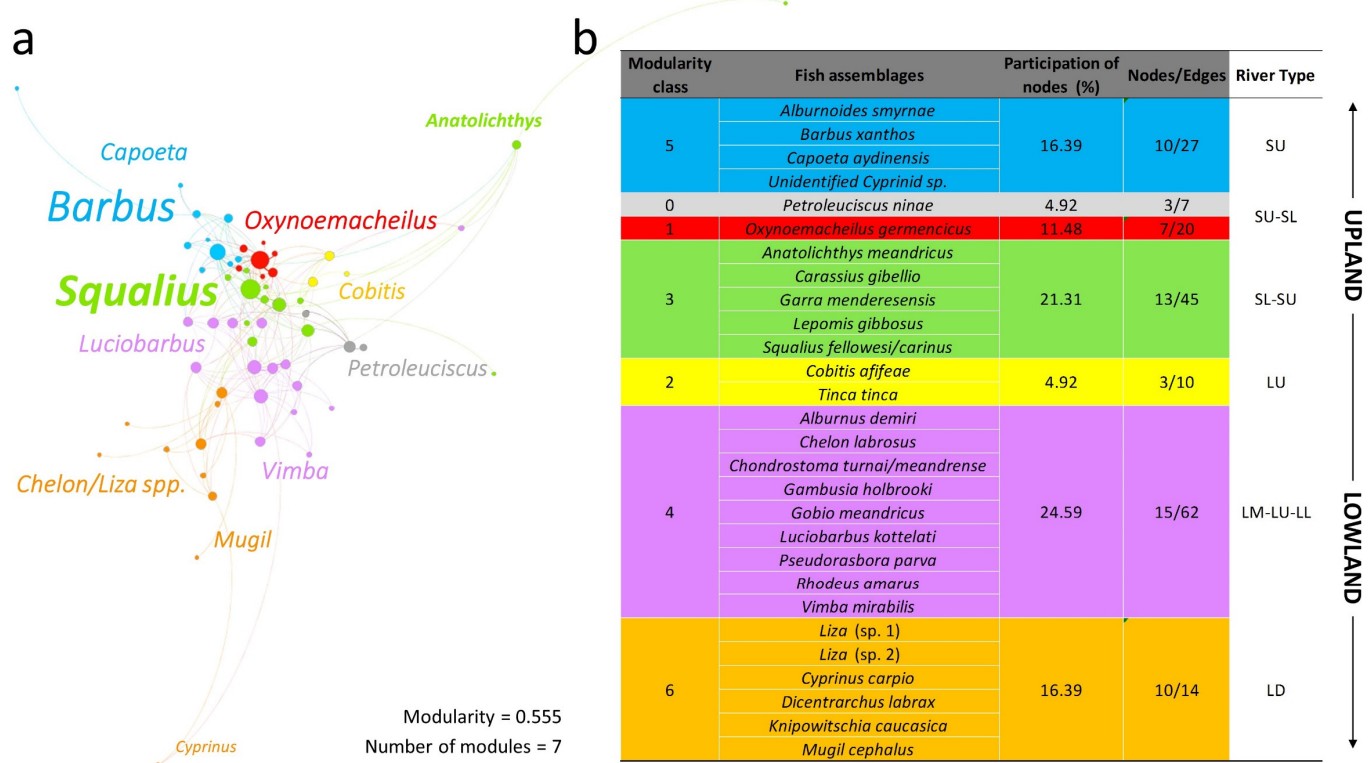

**Figure 7.** Bipartite network of fish assemblages based all samples: (**a**) different module colors display the classes (assemblages) derived from the modularity test (see text). The names of the assemblages are given in size by the most dominant fish genera (left); and (**b**) the main species in each modularity class are listed and each module is further compared to the biotic river types developed through the cluster analyses of selected reference samples (final column at right). The assemblage groupings and biotic river types are arranged in an upland–lowland procession.

In the Büyük Menderes, the adapted EFI+ application assessed most upland sites and samples as being in "good" condition. Many sites that were noticeably impacted by multiple anthropogenic pressures in the lowlands, the main stem, and above barriers to fish movement were classified as being in moderate condition, whereas only six samples were assessed as "poor". In a superficial sense, this initial screening did react to a real upland–lowland degradation gradient pattern, since the general degradation trend was diagnosed by EFI+. However, the spread and the strength of diagnosis were unacceptably narrow in scope, with no sampled sites in "high" status and only one in "bad" status.

In order to explore the EFI+ index application results, the site quality index (SQI) was employed for an initial comparison. Since the SQI was assessed independently at each site to express the relative anthropogenic pressures on the fishes and the river ecosystem, this application seemed proper. The overall correlation of the EFI+ classification per site with the site quality index was $R^2 = 0.37$ (Figure 9). There seemed to be noticeable consistency among the more degraded sites in the lower larger parts of the river (most were assessed as moderately degraded and below the "good" threshold). However, it seemed that the fish-based index did not respond well in low-species conditions. When low-species sites were deleted from a comparative subset, the correlation between EFI+ and SQI was much better at $R^2 = 0.56$ (Figure 9).

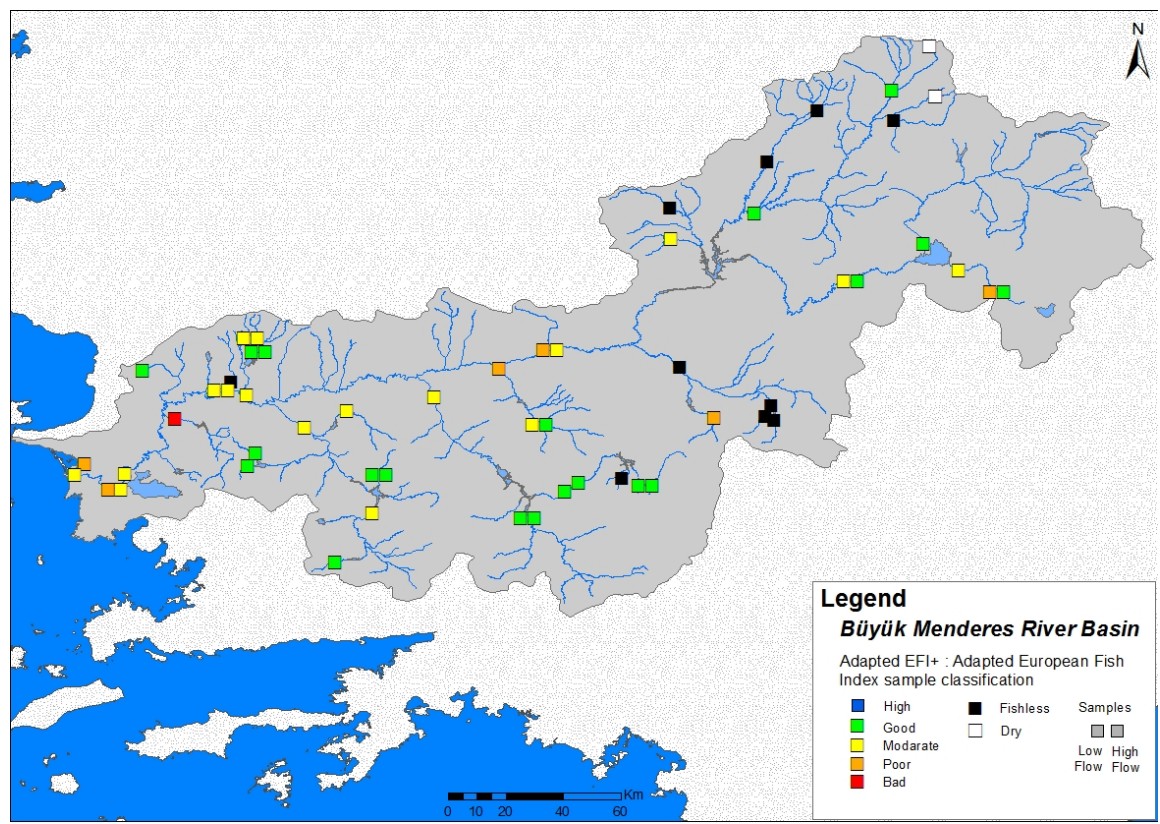

**Figure 8.** The European Fish Index (EFI+) classification results for each sample/site. Note that site results shown side-by-side refer to low-and-high flow results in eleven sites that were sampled twice (i.e. spring-summer). Sites labeled as "dry" may not have the ability to support fishes due to their intermittent flow regime, while all sites that were fishless had severe degradation (alteration/connectivity problems and/or severe pollution pressures) that certainly affect fish populations.

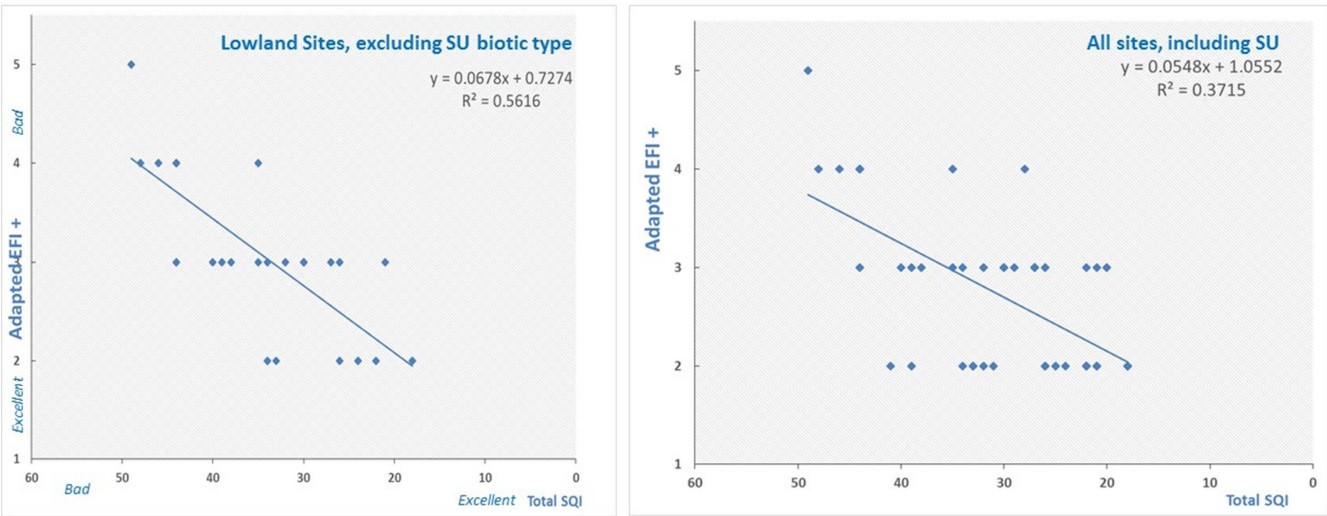

**Figure 9.** The adapted European Fish Index EFI+ (on Y-axis) correlated with the site-based assessment using the locally assessed site quality index. All sites and samples in all biotic types of rivers are provided on the right, and a subset of only lowland sites (excluding SU biotic type sites) is shown on the left. Due to a drawback in assessing low-species streams, the EFI+ bioassessment provided a better correlation in the species-rich sites.

## 4. Discussions

### 4.1. Achievements

To our knowledge, this survey is the first time a model-based assessment has been attempted for fish-based bioassessment in a river basin in Turkey. This work goes beyond water quality assessments [68] by utilizing fish assemblages to reflect the problem of multiple pressures on ecological integrity in rivers. Utilizing the tactical steps of EU WFD fish-based bioassessment, we constructed preliminary biotic river types for much of the basin. Classification analyses using the fish samples and abundance data per sample documented a biotically based typological framework for the first time in this basin. This was augmented with network clustering of the entire collected fish assemblage data at 44 river sites. Anthropogenic pressures were analyzed for each surveyed river site to provide background knowledge (pre-classification of degradation) in order to compare this with the fish-based index results. Lastly, the model-based EFI+, a fish index widely used in the EU, was slightly modified to accommodate local Anatolian species, replacing them with proxy European taxa of presumably ecologically equivalent function already embedded within the EFI+ model. The procedure of applying the fish-based index allowed us to explore fish as indicators according to the premises of ecological integrity as applied in the EU WFD procedure for rivers in Europe.

The Influence of anthropogenic impacts on riverine conditions has a complex geographical pattern in the Büyük Menderes. Beyond the fish index results (see below), our study made many empirical observations in relation to fish community and habitat degradation. It was obvious to the researchers during sampling that fish communities changed in most areas due to various forms of anthropogenic degradation, presumably often due to habitat degradation and connectivity loss (often with multiple pressures acting at different spatial scales). Although some fishes may have contracted in range or even become locally extirpated, we have little evidence that current fishing/fishery pressure has widespread negative effects. During field work, few fishers where encountered (an exception being the delta where a strong fishery is focused on marine-migrant fishes). Our work reveals many new questions with respect to the reasons behind the observed fish assemblages and the perceived fish population impoverishment.

On the basis of our observations and the relevance of observed multiple anthropogenic pressures, many examples of degraded fish community structure are evident in this survey. Although a fairly large basin, the site-level survey results showed a rather low species diversity (as compared to the total species pool known to be present). Many of the degraded sampled sites also had a predominance of small-sized fish in nearly all samples; that is, size classes <15 cm TL dominated the samples (see Table S1). Generalist species, such as eurytopic small-sized fishes that are tolerant of disturbed conditions, dominated in many river reaches of the Büyük Menderes. Specialized species that are adapted to complex river–floodplain conditions were very scarce compared to the widespread generalist species. This situation seemed to persist even where remnants of adequate microhabitats existed. As would be predicted in typical European temperate rivers, the alien species populations in the degraded parts of the river were rather high, especially in the lowlands and near artificial reservoirs or other artificially impounded waters. This trend has been observed in many parts of Turkey and may be increasing [69,70]. Some aliens, particularly *Gambusia* and *Lepomis*, seem particularly tolerant of disturbed conditions in this river system. Several native lacustrine and stagnophilous fishes seem to have become scarce, probably mainly as a result of floodplain degradation, morphological homogenization (channelization), and other stresses. This included habitat specialists, such as several globally threatened endemic species in the genera of *Garra*, *Pseudophoxinus*, *Cobitis*, and *Anatoloicthys*. These species had a very localized and/or range-restricted distribution both in our survey and in the surveys of Güçlü and colleagues [34]. Even in some upland larger sections of the river, otherwise widespread fishes such as *Alburnoides* and *Vimba* were unusually scarce. Some larger migratory fishes have probably become rarer because their lotic reproductive habitats are severely degraded, fragmented, or are unreachable (e.g., due to many artificial barriers).

Multiple anthropogenic pressures and landscape-scale changes have been documented in the basin [71]; thus, we considered our observations a result of serious degradation and presume that our sampling effort provided an adequate sampling baseline.

In fact, there is no doubt that much of the river has been extensively and severely degraded, especially in recent times. It is well known that, since the 1990s, the Büyük Menderes basin has had notorious water stress problems [72]. The reduction in river water flow also exacerbates seasonal pollution impacts. Furthermore, fish and in-stream biota conditions may also be affected by drought events, although the rainfall patterns were near-normal during the survey period, but showed variability among recent years [73]. It is important to consider that, in and immediately after drought periods, added stress would be evident on the fish assemblage statistics [74]. Pollutants can impact fishes in sublethal concentrations; since fishes are long-lived organisms, these impacts may take years to be expressed [75]. Severe pollution effects often increased by drought periods are probably an important cause of fish population decline, especially of the so-called intolerant species [21,76]. Moreover, some smaller tributaries have extremely polluted river segments, especially near industrial or urban complexes such as near Denizli. Some sites, such as those downstream of Denizli had no fish, presumably due primarily to pollution. Such industrial and urban-based pollution severity was also documented in earlier studies [77]. Recent pollution-related mass fish kills (evident in the local and national media) also support the notion of widespread assemblage/abundance impoverishment evident in our fish surveys in lowland areas of the study area.

### 4.2. Identifying Problems and Shortcomings

In applying bioassessment using fish as indicators, what is particularly challenging is comparing the survey results with reference baselines since there is scant knowledge of the area's local ichthyofauna before recent anthropogenic degradation. Modern anthropogenic pressures have widely altered conditions especially during the last 50 years in this river [24,25,78]. This challenge is not restricted to the Mediterranean basins of Turkey; these problems exist nearly wherever new indices are being developed [54,79]. Describing the reference conditions in relation to the fish community in different parts of this river basin and understanding the state of "high ecological integrity" per river type is a pivotal aspect of bioassessment and monitoring [8,80]. Ideally, an understanding of type-specific biological reference conditions should be based on the study of pristine sites without human-induced disturbance. Regrettably, no such sites exist in the main stem and most of the lower elevation tributaries of the Büyük Menderes. In the lower part of the river, the differences between today and the recent past are quite remarkable (Figure 10). We are led to presume that fish assemblage changes may have also been significant, in line with landscape-scale habitat changes.

There is a very limited research effort in terms of river ecosystem studies, particularly in the use of aquatic biological indicators in rivers in Turkey [25]. Several recent studies quantifying environmental and aquatic degradation exist, even for the Büyük Menderes [24,28,81,82]; however, studies that define fish communities and fish-based bioassessment are still extremely scarce in Turkey [25,80,83]. Lastly, this data-scarce situation supports the opinion of Ergonul and colleagues [19], who showed that the task of developing a country-wide fish-based index for rivers in Turkey is especially challenging due to the lack of knowledge on the ecology of fish species (especially endemic fish) and the absence of long-term datasets, with nearly no historical ecology study of the river changes.

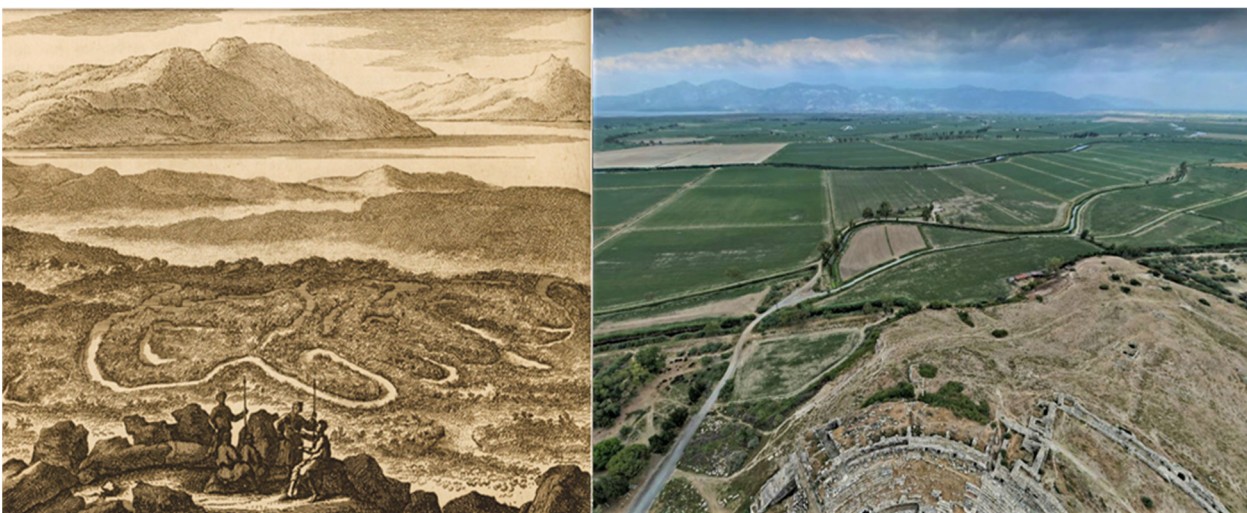

**Figure 10. Left**: Early 18th Century landscape painting of the lower valley with tortuous meandering and wooded riparian swamps (probably near Aydin) in 1714, by Cornelis de Bruyn (Public Domain/A. Laskaridis Foundation—Alexander S. Onassis Public Benefit Foundation). **Right**: Drone photo of a heavily modified section of the main stem of the river and drainage canals at Ancient Miletus, with the ruins of a classical theater in the forefront (by Hasan Urey).

Initiatives for building monitoring protocols and indices for fish-based bioassessment may help in promoting a scientific understanding of ecological integrity in river ecosystems [84]. Only through interdisciplinary research and long-term monitoring can the fish community dynamics be adequately described and the factors that influence fish communities in such disturbed conditions be fully appreciated [85,86]. The premise of using ecological integrity as an indicator framework is well respected [87,88] and is foundational for the EU WFD approach [7,79]. Baselines should be informed by broader history and natural history knowledge; otherwise, one may fall victim to the "shifting baseline syndrome", i.e., when human perceptions of altered conditions can misguide assessment and conservation management [89]. We were very cautious not to be entrapped by a shifting baseline syndrome in our study. This is a major reason why indices should be used with great care.

In the present study, EFI+ provided an assessment using model-based references associated with the sampled site's specific environmental parameters. Our application of a locally modified EFI+ was defined by two basic results: (a) the index loosely reflects fish assemblage degradation correlating with ecosystem quality (SQI); however, (b) the breadth of degradation indicated is very poorly depicted by the index results. The EFI+ provided a very narrow assessment range failing to accurately and consistently assess the severity of degradation. This last point is critical for the evaluation of the index's applicability. The fact that many severely degraded sites are assessed as "moderate" is a particularly serious shortfall of the index. In our study, the modified EFI+ failed to assess the full breadth of a five-scale classification gradient. The EFI+ assessment values rarely ascended above good or descended below moderate in this application. We would expect a fish-based index to define degradation more consistently, especially in such severely degraded conditions.

Although we did not attempt to further validate the index results, we may propose some possible shortcomings that could be relevant to the failure of the model-based index in our study. Several potentially useful metrics are not utilized in the EFI+. Our survey indicated that, as conditions of ecological integrity became more degraded, the prevalence of alien fishes increased, just like in other studies [90]. Migratory fishes seem to react importantly to degradation [54], and these are not taken explicitly into account in the EFI+. Where the river type naturally has a very low number of fishes (low species/low density population) such as in the uplands, the EFI+ is prone to misclassification. If the

very low density is caused by human impacts, the EFI+ manual recommends that "more simple methods or even expert judgment is sufficient to assess the ecological status of the river" [17]. One way of ameliorating the index is building within it more relevant metrics [80,91]. It might be worth investigating other applications of model-based fish indices in stressed situations to see how similar problems have been addressed.

Lastly, sampling rivers with non-wadable sections may not have been adequately achieved in this project in some instances (where deep-water sections dominate). Although we are confident the sampling effort was not an overriding issue in the failure of the index in this study, we believe that sampling difficulties in large rivers are a potential impediment worthy of further investigation.

*4.3. Insights and Recommendations*

Fish can be effective indicators of ecological quality, and they should continue to play an important role in monitoring [92], including new ways to track fish community assemblages [93] and of routinely using fish as indicators in bioassessment, conservation management, and restoration. The concept of ecological integrity has gained attention as one of the ultimate goals in nature conservation [94] and bioassessment in rivers [88]. Fishes are very important in guiding the study of the past conditions and future desired states in managed river ecosystems.

In our study, the adapted European Fish Index EFI+ was applied provisionally as an exploratory tool. It proved to be a very "blunt tool" without a biogeographically honed specialization needed for monitoring rivers in Anatolia. New fish-based indices must be constructed as seen in most countries in Europe in recent years [95]. We recommend a multimetric index be developed for the Western Anatolian Ecoregion, of which this basin is a part; both spatially-based approaches and model-based indices [54] should be further investigated. A holistic approach to bioassessment and monitoring frameworks is needed, and this will require substantial funding and concerted effort [37].

The EU WFD promotes an ecosystem approach to water management that may also accommodate conservation actions and a more holistic management framework which should include steps for restoration [96,97]. Scientists, local communities, and society must be engaged in this development. Moreover, aquatic ecosystem monitoring and biodiversity conservation must not be totally separate endeavors. In our study area, the requirements of the fish must be taken into account, both to understand the fish as environmental indicators and to be able to conserve and restore fish populations and species assemblages in a degraded and heavily modified river system. Many of Turkey's endemic species are threatened within a serious biodiversity crisis that especially affects freshwater ecosystems [98]. Despite all these conservation imperatives, fishes are often "out of sight and out of mind" when it comes to river basin management and conservation in Turkey, as in several other Mediterranean countries. The lessons gained from this exercise in the Büyük Menderes underscore the value of careful "whole-scape" approaches [99] that should include broader and in-depth natural history inventory and science-guided ecosystem-based research monitoring and conservation initiatives.

Experience in conservation in Europe and North America shows that the survival and recovery of many Mediterranean-climate river fish communities depends on the availability of high ecological integrity and permanent perennial flowing refuges, where longitudinal connectivity and persistent seminatural hydrological conditions are maintained. Protected areas are important, but they are not the only solution. Since no "protected area is an island", the procedure should follow the management of ecological basin units [84], i.e., prioritizing conservation actions within strategic basin segments. Strategies that combine conservation and water management must be investigated [100] especially in the face of resource uncertainties in the near future. As outlined in many recent reviews, global climate change is predicted to increase the frequency and magnitude of extreme weather conditions, thus further altering aspects of river habitat and fish communities [101], as well as threatening the survival of sensitive species [1]. Mediterranean freshwater ecosystems

are highly vulnerable to climatic change, mainly because of the limited dispersal abilities of most aquatic species. The direct impact of weather conditions on aquatic habitats, and the fact that many aquatic systems have already been severely impacted by other human activities may further degrade the natural structure of fish communities [102] and may lead to further species loss.

We recommend the following steps for further research for combining both ecosystem quality monitoring (i.e., EU WFD approaches) and effective biodiversity conservation initiatives in the Büyük Menderes:

- A complete taxonomic inventory must finally complete the ichthyofaunal natural history knowledge of the river basin. The mapping of all species distributions and habitats is critical for understanding fish communities;
- A historical study of species distributions and human-induced changes must be investigated. This includes careful analysis of the history of habitat changes including a socioecological research approaches (e.g., engaging fishers and local communities).
- Fish-based monitoring techniques and a long-term monitoring initiative must be set in place in order to explore trends and patterns of change. This new research program must be integrated within a basin-wide biodiversity conservation strategy.

Lastly, relevant conservation-science and research programs must aim to preserve and restore specific river and riparian types, specific habitat areas, fish communities, and fish species [13]. Long-term adaptive management studies that also focus on understanding the river processes, conditions, and trends should assist in effective restoration and conservation management actions [103].

## 5. Conclusions

This project explored how fish assemblages may express ecological integrity following the rationale of the EU Water Framework Directive, just beyond the biogeographical boundaries of the European continent, in a major Anatolian river basin. Anthropogenic changes have affected fishes in most river water bodies of the Büyük Menderes basin, and this was diagnosed both by a pre-classification of anthropogenic degradation (SQI) and by the trends shown in the adapted version of the model-based European Fish Index used in this study. EFI+, however, failed to adequately assess the extreme degree of severity and degradation in many of the impacted river sites. These results provide insights for further work in this important research area. Ecological research and monitoring procedures such as fish-based indices constitute tightly interconnected scientific fields and should be integrated to achieve better tools for environmental assessment and conservation planning. Long-term research and commitment to conservation is especially important in Turkey because, unlike most temperate European areas, the inland waters of Anatolia host a high percentage of local endemic species, many of which are severely threatened and scarce within the river systems [34]. Running waters are now considered among the most endangered of all natural ecosystems in Turkey [22], with biodiversity loss representing a major threat to their structure and functioning.

**Supplementary Materials:** The following supporting information can be downloaded at: https://www.mdpi.com/article/10.3390/w15122292/s1, Table S1: Data collected for the application of the EFI+ index for each site.; Table S2: The European Fish Index (EFI+) adaptation of the species.

**Author Contributions:** Conceptualization, S.Z., N.K., S.C.Ő. and Y.C.; methodology, S.Z. and N.K.; software and formal analysis, N.K., S.Z. and Y.C.; investigation, S.Z., K.Y., V.V., P.G.K. and S.C.Ő.; resources, S.C.Ő. and G.K.A.; data curation, N.K. and S.Z.; writing—original draft preparation, S.Z. and N.K.; writing—review and editing, S.Z., N.K., V.V., S.C.Ő. and G.K.A.; supervision, S.C.Ő.; project administration, Y.C.; funding acquisition, Y.C. All authors have read and agreed to the published version of the manuscript.

**Funding:** Initial field research was funded through the European Union within the EuropeAid project "Technical Assistance for Capacity Building on Water Quality Monitoring" EuropeAid/131199/D/

SER/TR. The first author wishes to thank Alcibiades N. Economou (HCMR) for permitting involvement in this research while the author was on a limited day leave from the institute.

**Institutional Review Board Statement:** All ichthyological field surveys providing data for the manuscript entitled "Fish assemblages as ecological indicators in the Büyük Menderes (GreatMeander) River, Turkey" were conducted in accordance with Turkish law, agreements among participating institutions and conformed to ASAB/ABS ethical guidelines. The ichthyological team which Stamatis Zogaris and Saniye Cevher Őzeren led as principal investigators, along with the University of Ankara had secured all necessary permits and permissions under contract with the EuropeAid-funded expedition. This is made clear in the Manuscript of the paper. It is important to note that during the time of the surveys (2013–2014), an Animal Welfare Committee was not a requirement in the country of research or within the said University; therefore, obtaining a specific ethical license to conduct the field samplings was not a requirement and was not possible at the time.

**Data Availability Statement:** Raw data is available from the corresponding author.

**Acknowledgments:** The authors greatly appreciate the assistance during field work of Aris Vidalis and Dimitris Zogaris, who participated in sampling at many sites. The Pammukalle University staff, especially M. Duran, assisted greatly in many aspects of organization. The authors are especially grateful to Salim Serkan Güçlü for a thorough review of our species list and his taxonomical corrections, and to Jörg Freyhof with identification help in an early part of this project. Hasan Urey is thanked for a drone photograph published here. Lastly, the authors would like to thank the staff of ENVECO Ltd. for help within the project "Technical Assistance for Capacity Building on Water Quality Monitoring" EuropeAid/131199/D/SER/TR.

**Conflicts of Interest:** The authors declare no conflict of interest. The funding agency that facilitated this work had no role in the design of this paper; in the collection, analyses, or interpretation of data; in the writing of the manuscript, or in the decision to publish the results.

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
