# Peer review of "Fish Assemblages as Ecological Indicators in the Büyük Menderes (Great Meander) River, Turkey"

_water, doi:10.3390/w15122292_

Round 1
Reviewer 1 Report
MS: Water-2350164
Zogaris, S. et al.
Title: Fish assemblages as ecological indicators in the Büyük Menderes (Great Meander) River, Turkey
Referee’s Report
This is a very interesting paper, which in many respects is a first attempt to tackle a situation about which relatively little is known. A lot of work has gone into this both in terms of the field effort and the data collection and analysis.
I found the ms easy to read, with only the occasional English infelicity, and the Table and Figs easy to read, although I would ask for a little more explanation of e.g. Table headings in the legend. The quality in my copy was also a little uneven (e.g. Fig 3 appeared slightly fuzzy), but this may be sorted in the final printed version.
I did find both the Introduction and the closing Discussion slightly wordy (see e.g. some of the comments below) and I think it would be worth the authors’ trying to tighten them up.
I append some more detailed comments below.
Detailed comments:
|
Line |
Comment |
|
21 -7 |
It is not clear whether the EFI+ was applied only to the less-degraded or to all collections. Please clarify. |
|
55-7 |
This largely repeats what’s already been said in lines 47-50. |
|
66 |
‘staggered” ? Perhaps ‘lagged’ ? |
|
151 |
‘spp’ non-italic |
|
156-8 |
Unclear whether statements or reproduction (or both) in 2013-14. Are you suggesting this was an exceptional year? Clarify |
|
308 |
For pollution was this based solely on visual clues (outfalls?) or was bibliographic information (e.g. Ref 26) taken into account as well? Is there an accepted methodology to which you can refer? |
|
319 |
. . . . trait categories . . . . |
|
333 |
. . . . endemic and range-restricted species . . . . |
|
433 |
It would be helpful to divide this also by flow (high/low). Presumably 2 samples indicates h/l ? Are there differences with season (h/l)? |
|
443-8 |
I am not clear how the ‘best available’ were selected. Are these intended to represent a ‘baseline’ or ‘reference’ state? If so, how can ‘severely degraded’ sites be included?
I think you also need a better basis for the category limits than ‘selected’ (how?) step changes. What values have previous workers used/suggested? |
|
Fig 7 |
I am curious as to why Anatolichthys appears out on its own, yet is assigned to the SL-SU grouping, despite being graphically closer to the SU-SL and LU groupings. You might think to insert a line of explanation. |
|
538 |
. . .no sites . . . ? |
|
540 Fig 9 |
I think you would be better using the actual EFI+ scores rather than the (very much broader) categories and graphing one continuum against the other. From this correlation, how do your SQI limits measure out (see comments 443-8)? |
|
591 |
‘we’ ?? |
|
590-1 |
Is fishing a pressure? See also line 616 and species such as Acipenser |
|
622-3 |
This assertion is not well-supported here: more data/refs needed. |
|
622 |
How does this meet your categories in Fig 3? See also comments lines 443-8 above. |
|
643 |
Again see comments lines 443-8. It might be worth looking at other applications of the EFI in stressed situations to see how similar problems have been addressed. See e.g. https://muse.jhu.edu/pub/423/article/809435/summary in estuaries. |
|
762 |
This is a very important recommendation and could usefully go into the Abstract (which may be all some people read!). |
|
800-13 |
Beware over-saturation of recommendations! You might consider it stronger just to go with one simple strong message (line 762?). |
I found the ms easy to read, with only the occasional English infelicity
Reviewer 2 Report
Comments to Authors
General comments
The manuscript presents interesting and accurate data aimed at eliciting freshwater fish assemblages as ecological river-quality indicators of the Büyük Menderes River system in Turkey. The work also stresses the partial unsuitability of the IFI+ index to represent ecological status of river systems in endemic fish species-rich regions, such as Anatolia. The methods adopted (both sampling and statistic elaboration of the data) are convincing and sufficiently described, although I have one major question about some of work’s methods. The sampling operations are dated, as they were carried out ten years ago. This has implications on the representation of the data, which actually describe the different ecological status displayed by fish assemblages across river sites in response to different anthropogenic impacts at that time. Therefore, my main question is which data had been used in SQI calculation, beside direct visual inspections and assessments. For instance, are remote sensing environmental data and bibliographic information used coeval with sampling operations? They might display huge differences between 2013-2014 and 2023 situations because of change in 1) land use and/or alteration due to development of civil engineering and so on; 2) availability of literature information due scientific updating occurred in a 10 years period. Not coeval data would impair the relationship the authors investigated between the IFI+ and SQI indices, with consequences on some of the main findings of the paper. To ultimately solve this issue, I think authors should clearly indicate references to data sources, of both remote sensing and literature information, in the SQI-dedicated section in methods.
I think another important problem is the English language used, as I noticed great differences in style and correctness across the different sections of the manuscript. Some parts are well and correctly written, while some other have problems in grammar (grammatical errors, verb tense, punctuation) and style (such as verbose sentences and sentence construction) (see specific comments). Therefore, I recommend the ms to be checked by a proof-reader or a native English-speaking colleague to improve and uniform the quality of the language.
Overall, I recommend this work to be published in Water, after major revisions.
Specific comments
Abstract
Line 18 and 22: please avoid semicolons when possible. Rather, link the sentences
Line 23-27: Unclear, please rewrite these couple of sentences
Introduction
Line 33: why not referring also to “lotic” ecosystems. There are also upland sites in you sample, that is waters displaying strong currents due to the abrupt variation in altitude (lotic waters).
Line 38-41: please rewrite in a streamlined manner
Line 46: I think the word “longitudinal” is improper here when describing the general pattern of variation in the fish assemblage of a river. The gradient is simply observed between the spring and the delta regions of a river, while longitudinal refers only to rivers flowing some like E-W or W-E direction, such as the Büyük Menderes River system in Turkey. Please rearrange the sentence
Line 47-50: too much verbose, please simplify the sentence.
Line 52: please avoid semicolon here and throughout the whole manuscript when possible.
Line 62-63: “ and ecological….applied.”???
Line 78-81: please simplify and streamline the sentence.
Line 85-88: Too long and complicated sentence. Please divide the period and simplify
Materials and Methods
Chapter 2.1: I think this section provides information that is midway between introduction and method. If the information reported was used as a method (for instance some cited literature used to calculate the weight of some of the criteria to obtain the SQI), then a tabular form may be preferred in my opinion. I think the authors should move some parts in the introduction and schematize the other parts in a more condensed form i.e. a table.
Line 162: Check “then”
Lines 256-259: Very intricate sentence. Please try to explicit concepts
Line 275-276: The authors devoted the proper effort for this part that is very important. However, ultimately I did not understand the method used to establish the reference conditions. How did the authors assign the reference conditions of the river sites investigated? The sentence appear trunked and information provided unclear. Please add a clear statement on how you established the reference conditions
Line 281: add a reference to remote sensing data (date and source of acquisition) and specific literature information used)
Line 299-300: Which are the thresholds for grouping sites into the three-category degree of degradation (minimal, slight, and severe)? Distribution quartiles? Tertiles? Please add the criterion you used to group sites into degradation categories.
Lines 300-303: This part is some like circular and maybe not useful for comprehension. Are there differences between the global degradation index and the IFI+? In other words, are author using the IFI+ as a global degradation index? If I understood correctly, the authors wanted simply to stress the idea of correlating the SQI with the modified IFI+ to check for the suitability of the latter in describing the river ecological status in an endemic species-rich region. As this is the important message to be communicated, I suggest the authors to split this long sentence making clear what the authors are referring to.
Line 321-323: Please take into consideration the comments made on lines 275-276 and on line 281
Lines 323-328: Again, are the environmental site data coeval with sampling operations?
Lines 329-350: The authors attempted to adapt IFI+ to the river studied and the framework adopted is convincing, although the method can have some questionability. Species that are endemic show unique adaptations (morphological, ecological and so on) because they experimented a unique environment. Therefore, the method can have a sort of circular argument since I presume you have endemic species because environmental features of Anatolian rivers are ultimately not so similar to the western Mediterranean rivers (IFI+ is based upon 12 environmental descriptors, which can strongly vary between the two regions). However, this exercise is an attempt and the ecological equivalence of species and the consequent fish taxonomic adaptation used is interesting and it can be accepted taking into account the limitations before described.
Line 350-379: The absence of Salmo type species in the river system studied is in itself emblematic of degradation. It impeded the use of the complete IFI+ and limited its use to Ciprinid type species. Therefore, the question is: Are undisturbed sites really present in the study area? However, I understand that pristine reference conditions are far to be still present nowadays worldwide.
Results
Line 382-384: these are methods. Please move in this section
Chapter 3.1: The text is a mix of results, methods and discussion. Please refer only to results and move the other information in the proper sections. Please remember that usually references cannot appear in a result section.
Chapter 3.2: please remember comments on thresholds for defining the three general categories of degradation (the word “arbitrarily” in fig 3 can produce questionability of the results)
Figure 8: I did not manage to notice the difference between sites defined as “low flow” and “high flow” samples in the map site labels. Corresponding figure labels appear identical in the legend on the left of the figure.
Chapter 3.4: The text appears as written following a discussion style (lines 535-539 and 546-548). Move some of the considerations to discussion and maintain text adherent to results only.
Discussion
Lines 566-569: There is a little bit of contradiction in this text
Line 583. Add the limitations of the method and potential drawbacks (see previous comments)
Lines 592-593: Is it necessary?
Conclusions
Try to summarize and simplify the text, it is too long for a conclusion section.
English revision is recommended
Reviewer 3 Report
· The objective of the manuscript is not clearly established. Authors said “we survey river ichthyofaunal assemblages using standardized sampling techniques and describe preliminary steps…to explore the patterns and trends in distribution and potential effects of anthropogenic pressures on fish assemblages…” Here, it is confusing indeed. Which metrics of fish assemblages are considered? Just distribution? How distribution can be related to anthropogenic pressures? Which pressures? Which geographic region was the surveyed conducted?
· In Methods there is not a clear statement on how the survey design was implemented to sample fish. There are just explanations of when the sampling was done and that there were sampling campaigns in 37 seven rivers with 44 sites, but there is not any justification of selection of sites based on any expected comparisons of habitat or particular characteristics for instance from close to far of any human settlement as potential human influence.
· The work is not clearly established in design as to surrogate fish assemblages as ecological indicators and if they were so, there is not any idea which ecological indicators they would be.
· The assessment of anthropogenic pressures is subjective since authors used visual inspection, remote sensing and bibliographic references and authors used a value from 1 to 5 to assign a condition. However, it is not clear if these refer to quality of the area or any pressing factor. Authors say something like an index, but it is not clear.
· It is not clear if the Site Quality Index (SQI) was something authors implemented for the work itself or it was an index used in previous works.
· Authors refer to a European Fish Index (EFI+), which is a model-based index using site-specific reference values in order to calculate reference condition baselines. However, authors do not clearly explain how this model is going to be used incorporating data from the study.
· Consequently, the title of the work went short in relation to the content since nothing is mentioned of the model in title or even in the objective if this model is being the core of the work.
· In general, the amount of results obtained is not reflected in objective nor in title.
· In the end, was it a work to identify trends in fish assemblages or to identify mostly the condition of given habitats?
· Which is the implication of having found what were the ten most common fish species in the work? Are they under risk of extinction according to IUCN’s Red List? Are they commercially important?
· Table 4, for instance, comprising the species collected per water body, is relevant to show how many species are there? Specify what FISHLESS means for the purpose of the Table. I infer without any fish.
· In the first sentence of the Discussion authors claim “…study charts fish assemblages in relation to environmental degradation..”. However, it is not clear how. Then, authors claim “…this survey is the first time a model-based fish-based indicator assessment has been attempted utilizing an extensive standardized sampling campaign..” However, it is not clear what authors wanted to make clear here. Then, authors said “The ichthyofaunal list compiled in this study is similar to a published distributional study done in recent years”. However, authors just claim differences in taxonomy instead on other matters.
· I do not consider viable to “construct preliminary biotic river types…hypothesis-led approach for fish-assemblage reference baselines”.
· Authors claim that “Our work provides specific empirical evidence for assemblage degradation and also reveals many new questions…” There was not any evidence for assemblage degradation.
· There are many arguments in the Discussion that are not supported by the evidences in the Results.
· Conclusions are not align with apparent key results in the study design, if any.
The work is saturated with a lot of analysis but the backbone is not clearly established from the very beginning. As it stand, the manuscript requires a substantial amount of work and reestructure.
Round 2
Reviewer 1 Report

Fine: perhaps minor editing
Author Response
See file submitted

Reviewer 2 Report
The revised version of the ms addressed convincingly all the comments I made-
Therefore, the work can be accepted in the present form
Author Response
We would like to thank the reviewer for his/her valuable comments.